# Rethinking Complex Queries on Knowledge Graphs with Neural Link Predictors

**Hang Yin**
Department of Mathematical Sciences
Tsinghua University
`h-yin20@mails.tsinghua.edu.cn`

**Zihao Wang**
Department of CSE
HKUST
`zwanggc@cse.ust.hk`

**Yangqiu Song**
Department of CSE
HKUST
`yqsong@cse.ust.hk`

## Abstract

Reasoning on knowledge graphs is a challenging task because it utilizes observed information to predict the missing one. Particularly, answering complex queries based on first-order logic is one of the crucial tasks to verify learning to reason abilities for generalization and composition. Recently, the prevailing method is query embedding which learns the embedding of a set of entities and treats logic operations as set operations and has shown great empirical success. Though there has been much research following the same formulation, many of its claims lack a formal and systematic inspection. In this paper, we rethink this formulation and justify many of the previous claims by characterizing the scope of queries investigated previously and precisely identifying the gap between its formulation and its goal, as well as providing complexity analysis for the currently investigated queries. Moreover, we develop a new dataset containing ten new types of queries with features that have never been considered and therefore can provide a thorough investigation of complex queries. Finally, we propose a new neural-symbolic method, Fuzzy Inference with Truth value (FIT), where we equip the neural link predictors with fuzzy logic theory to support end-to-end learning using complex queries with provable reasoning capability. Empirical results show that our method outperforms previous methods significantly in the new dataset and also surpasses previous methods in the existing dataset at the same time.

## 1 Introduction

Knowledge graph (KG) is a mighty knowledge base that encodes relational knowledge into a graph representation. However, due to the fact that modern knowledge graphs are often auto-generated (Toutanova & Chen, 2015) or constructed by crowd-sourcing (Vrandečić & Krötzsch, 2014), they are considered noisy and incomplete, which is also known as the Open World Assumption (OWA) (Libkin & Sirangelo, 2009). Complex query answering (CQA) on knowledge graphs is a practical task that can support many applications (Ren et al., 2023a; Wang et al., 2022). The CQA task requires answering the existential first order logic formula, involving logical operators, conjunction ($\wedge$), disjunction ($\vee$), and negation ($\neg$), as well as the existential quantifier ($\exists$). In particular, as CQA is based on KGs with OWA, it should perform reasoning, which utilizes available knowledge to predict the missing one where traditional traversal methods are doomed to fail (Ren et al., 2020).

To tackle this challenge, the query embedding method has been proposed (Hamilton et al., 2018), which aims to represent a set of entities by a low dimensional embedding. In addition to that, the logical formula is represented in an operator tree form (Ren et al., 2020; Wang et al., 2021), in which the logic operations are replaced with corresponding set operations. Especially, the existential quantifier induces a new set operation, set projection, which corresponds to the logic skolemization (Luus et al., 2021). Though there has been a line of research (Choudhary et al., 2021; Bai et al., 2022; Yang et al.,

2022; Wang et al., 2023a; Hu et al., 2024) following the same approach, this kind of methodology still lacks detailed inspection of its logic soundness or model expressiveness. Moreover, despite the operator tree form pushing a strict constraint on the solvable formulas, the real scope of the solvable logic formulas has never been estimated, and nowadays datasets are therefore highly restricted. In general, not much theoretical progress has been made to inspect on nowadays CQA models.

In this paper, we first review the inherent drawbacks of the existing prevailing operator tree form representation of query embedding approaches and clearly characterize the query family that has been investigated as Tree-Form (TF) queries. Then we extend our scope of research to the whole family of *Existential First Order queries with a single free variable* (EFO$_1$). Particularly, we represent queries as general multigraphs. Then we develop a new dataset containing ten new formulas which can not be represented in prevailing frameworks. Along with that, we propose a simple yet empirically effective algorithm which combines neural link predictors with strict fuzzy logic definition, and thus has strong theoretical guarantees. The algorithm is able to systematically infer EFO$_1$ queries of arbitrary complexity given any neural link predictor. Finally, we show that our algorithm is able to outperform existing methods in both our newly developed dataset and existing dataset. Our code and data can be found at `https://github.com/HKUST-KnowComp/FIT`.

## 2 PRELIMINARY

### 2.1 KNOWLEDGE GRAPHS

Given a set of entities $\mathcal{E}$ and a set of relations $\mathcal{R}$, a knowledge graph $\mathcal{G}$ encapsulates real-world knowledge as a collection of factual triples $\mathcal{G} = \{(a_i, r_i, b_i)\}$, in each triple, $a_i, b_i \in \mathcal{E}$ is the head entity and tail entity correspondingly, and $r_i \in \mathcal{R}$ is the relation among them. Based on OWA, the **observed** knowledge graph $\mathcal{G}_o$ is only part of the **complete** knowledge graph $\mathcal{G}$, meaning $\mathcal{G}_o \subsetneq \mathcal{G}$.

### 2.2 EFO$_1$ QUERIES AND ANSWERS

Existing discussions emphasize logical queries without universal quantifiers (Ren & Leskovec, 2020). Such queries are formally defined as Existential First Order queries with a single free variable (EFO$_1$) under the strict first-order logic theory. We provide a minimum set of definitions to characterize the EFO$_1$ queries on knowledge graphs following standard logic theory (Marker, 2006).

**Definition 1** (Terms). *A term is either a variable $x$ or an entity $a \in \mathcal{E}$.*

**Definition 2** (Atomic Formula). *An atomic formula is of form $\phi = r(h, t)$, $r \in \mathcal{R}$, $h$ and $t$ are terms.*

**Definition 3** (Existential First Order Formula). *The set of the existential formulas is the smallest set $\Phi$ that satisfies the following property:*

*(i) For atomic formula $r(a, b)$, itself and its negation $r(a, b), \neg r(a, b) \in \Phi$*
*(ii) If $\phi, \psi \in \Phi$, then $(\phi \wedge \psi), (\phi \vee \psi) \in \Phi$*
*(iii) If $\phi \in \Phi$ and $x_i$ is any variable, then $\exists x_i \phi \in \Phi$.*

We say a variable $y$ is bounded if there is an associated quantifier, otherwise, it is free. We use $\phi(y)$ to indicate the formula $\phi$ contains a free variable $y$. Then we finish the definition of the EFO$_1$ formula.

**Remark 4** (Query and sentence). *When a formula contains at least one free variable, it is also called a query. Otherwise, the formula can be called a sentence.*

**Definition 5** (Substitution). *For a EFO$_1$ formula $\phi(y)$, for any entity $a \in \mathcal{E}$, we write $\phi(a/y)$ or simply $\phi(a)$, for the result of simultaneously replacing all free occurrence of $y$ in $\phi$ by $a$.*

Then, we are ready to define EFO$_1$ queries and the answer set.

**Definition 6** (The Answer Set of EFO$_1$ Query). *The answer set of an EFO$_1$ query is defined by*

$$\mathcal{A}[\phi(y)] = \{a \in \mathcal{E} | \ \phi(a) \text{ is True}\} \tag{1}$$

All EFO$_1$ formulas can be converted to Disjunctive Normal Form (DNF), specifically:

**Definition 7** (Disjunctive Normal Form). *The disjunctive normal form $\phi_{\text{DNF}}$ of an EFO$_1$ formula is*

$$\phi_{\text{DNF}}(y) = \gamma_1(y) \vee \cdots \vee \gamma_m(y), \tag{2}$$

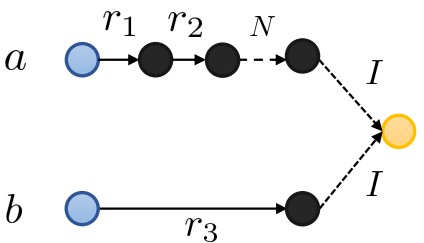

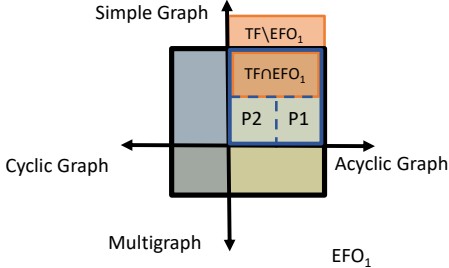

Figure 1: Representation of the tree form query "pni". We note that this kind of representation requires explicit set operators in the graph, the corresponding lines are dotted.

Figure 2: A diagram for the differences between the Tree-Form queries (orange blocks) and the $\text{EFO}_1$ queries (black box). $\text{EFO}_1$ queries are categorized by their query graphs. Some Tree-Form queries are not of $\text{EFO}_1$.

*where $\gamma_j(y) = \exists x_1, \cdots, x_k . \alpha_{j1} \wedge \cdots \wedge \alpha_{jn_j}$, $j = 1, ..., m$, where $x_i$, $i = 1, ..., k$ are existential variables, $y$ is the single free variable, $\alpha_j$. are atomic formulas or the negation of atomic formulas.*

There are many ways to represent a $\text{EFO}_1$ formula, however, the DNF has gained dominant priority in lines of previous research (Ren & Leskovec, 2020; Wang et al., 2021; Zhang et al., 2021). Converting the original query $\phi(y)$ into a DNF query $\phi_{\text{DNF}}(y)$ decomposes the answer set $\mathcal{A}[\phi_{\text{DNF}}(y)]$ into the union of the answer sets of $\mathcal{A}[\gamma_j(y)]$, which has become a common practice in recent research (Wang et al., 2023b). We consider $\text{EFO}_1$ queries in DNF throughout this paper.

## 3    THE LIMITATION OF PREVIOUS FORMULATION

In this section, we aim to answer the question: Can general $\text{EFO}_1$ queries be precisely represented by previous formulations in syntax? In recent years, a huge number of research (Ren & Leskovec, 2020; Wang et al., 2021; Zhang et al., 2021; Ren et al., 2023b) have all made such a strong claim. However, derived from the design of the previous method, we are able to characterize the query family that has been investigated as Tree-Form (TF) queries and provide it with a formal definition. Our discussion shows that the TF query family is not even a subset of $\text{EFO}_1$. Therefore, our discussion justifies the limitation of previous works by showing their unrigorous formulation.

### 3.1    THE SYNTACTICAL CLOSURE OF PREVIOUS FORMULATION: TREE-FORM QUERIES

The query types in the existing datasets (Ren & Leskovec, 2020; Wang et al., 2021), though targeted to $\text{EFO}_1$ query family, are selected with *bias* when it comes to the empirical evaluation. The reason is that the dominating way of addressing logical queries is to simulate logical reasoning as the execution of set operators on an **operator tree** (Wang et al., 2021; Xu et al., 2022), where each node represents a set of entities corresponding to the answer set of a sub-query. The logical connectives are transformed into operator nodes for set projections, intersection, union, and complement (Wang et al., 2021). Particularly, the set projections are derived from the Skolemization of predicates (Luus et al., 2021).

We characterize the expressiveness of operator tree method by providing the formal definition of the TF query, instead of confusing them with $\text{EFO}_1$ queries as the previous practices (Wang et al., 2021).

**Definition 8** (Tree-Form Query). *The set of the Tree-Form queries is the smallest set $\Phi_{\text{TF}}$ such that:*

*(i) If $\phi(y) = r(a, y)$, where $a \in \mathcal{E}$, then $\phi(y) \in \Phi_{\text{TF}}$;*
*(ii) If $\phi(y) \in \Phi_{\text{TF}}$, $\neg\phi(y) \in \Phi_{\text{TF}}$;*
*(iii) If $\phi(y), \psi(y) \in \Phi_{\text{TF}}$, then $(\phi \wedge \psi)(y) \in \Phi_{\text{TF}}$ and $(\phi \vee \psi)(y) \in \Phi_{\text{TF}}$;*
*(iv) If $\phi(y) \in \Phi_{\text{TF}}$ and $y'$ is any variable, then $\psi(y') = \exists y. r(y, y') \wedge \phi(y) \in \Phi_{\text{TF}}$.*

Figure 1 is an example of a Tree-Form query that appeared in the widely used dataset (Ren & Leskovec, 2020). The formal derivation of the definition of TF can be found in the Appendix B.

However, this biased selection of query types **deviates** from the original goal of complex query answering (Ren & Leskovec, 2020). It imposes strong assumptions for the formula (Wang et al., 2021), so a large part of first-order queries, even only with existential quantifiers, is ignored in both existing datasets and the solving strategies. Moreover, it is questionable whether existing query types

indeed skip the universal quantifier. As we have pointed out in the following, the recursive definition may seem to neglect the universal quantifier but fails when the formula becomes more complex.

## 3.2 TF QUERIES DEVIATE FROM THE GOAL OF $\mathrm{EFO}_1$ QUERIES

**Proposition 9.** *In DNF[1], the universal quantifier may exist in* TF *queries, thus the family of* TF *query is not a subset of* $\mathrm{EFO}_1$.

Proof: We derive this proposition based on the Definition 8:

$$r_1(a, x) \in \Phi_{\mathrm{TF}}$$
$$\exists x.r_1(a, x) \wedge r_2(x, y) \in \Phi_{\mathrm{TF}}$$
$$\neg\exists x.r_1(a, x) \wedge r_2(x, y) \in \Phi_{\mathrm{TF}}$$
$$\forall x.\neg r_1(a, x) \vee \neg r_2(x, y) \in \Phi_{\mathrm{TF}}$$

We note that in Definition 3, the negation is defined only on the atomic formulas, while in Definition 8, the negation is defined on the whole formula, which leads to the occurrence of the universal quantifier.

**Example 10.** *Based on the above discussion, the original "pni" query proposed in Ren & Leskovec (2020) and shown in Figure 1, should have this kind of formulation in DNF ($a, b \in \mathcal{E}$):*

$$\forall x.(r_3(b, y) \wedge \neg r_2(x, y)) \vee (r_3(b, y) \wedge \neg r_1(a, x)) \tag{3}$$

By proposition 9 and Example 10, we clearly show that both the methodology and dataset in previous research (Ren & Leskovec, 2020) deviate from its original ambitious goal to answer $\mathrm{EFO}_1$ query.

## 4 GAPS BETWEEN TF AND $\mathrm{EFO}_1$ QUERY

In this section, we aim to further identify the gap between the $\mathrm{EFO}_1$ query family and TF family. To this end, we first introduce the representation method for general $\mathrm{EFO}_1$ query.

### 4.1 A GRAPH-THEORETICAL DESCRIPTION OF $\mathrm{EFO}_1$ QUERIES

We discuss $\mathrm{EFO}_1$ queries by considering each conjunctive query $\gamma_j$ as a query graph. Graphs representation is already discussed in answering Tree-Form queries without negation (Daza & Cochez, 2020; Liu et al., 2022) and Tree-Form queries (Wang et al., 2023b).

**Definition 11** (Query Graph). *Let $\gamma$ be a conjunctive formula in equation 2, its query graph $G(\gamma) = \{(h, r, t, \{T/F\})\}$, each quadruple corresponds to an atomic formula or its negation, representing an edge with two endpoints $h, t$, and two attributes $r$, T/F, the relation and whether it is positive.*

We note nodes of different types represent the node to be a constant entity, existential variable, or free variable. Logical conjunction is naturally presented in the query graph because the order of existentially quantified variables is exchangeable, indicating the main strength of the query graph representation. Then, all $\mathrm{EFO}_1$ queries in DNF can be represented by a set of query graphs $\{G(\gamma_j)\}_{j=1}^m$. We show an example of query graph in Figure 3.

The concept of query graph is also similar to the Gaifman graph (Vardi, 2000), which is well investigated in constraint programming. The key differences are (1) query graphs emphasize more node types for each term and (2) query graphs also encode logical negation.

### 4.2 SYNTATICAL GAP BETWEEN TF AND $\mathrm{EFO}_1$ QUERIES

Figure 2 briefly categorizes $\mathrm{EFO}_1$ queries by whether their query graphs are acyclic or simple. We note previous research shows that Tree-Form queries are assumed to be both acyclic and simple (Hamilton et al., 2018). Then, the key to understanding the relationship between TF queries and $\mathrm{EFO}_1$ queries is to determine the gap between TF queries without a universal quantifier (the bottom half orange box) and $\mathrm{EFO}_1$ queries with an acyclic simple query graph (the top-right of the dark box). We characterize this gap by the following two properties.

---

[1]More generally, this conclusion is true in every prenex normal form, which is a pre-requisite for query embedding method (Wang et al., 2021).

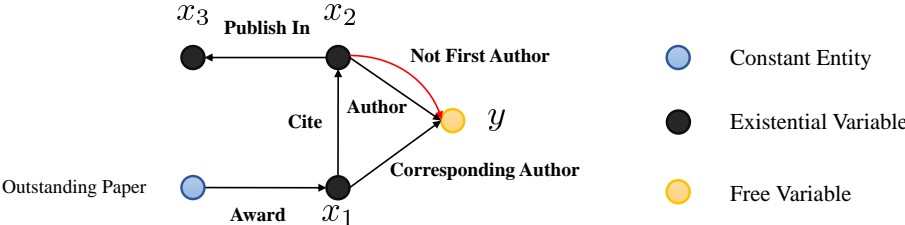

Figure 3: A given query "Find someone who has such two papers: s/he is the corresponding author of the first paper which has been awarded Outstanding Paper, s/he is the author but not the first author of the second paper which has been published and been cited by the first paper." can be represented as such query graphs. The formal existential formula is also given at the top.

**Property 12** (Negation without Constants)**.** *There is a negated atomic formula of the form $\neg r(x_1, x_2)$ or $\neg r(x_1, y)$ where $x_1, x_2$ are existential variables and $y$ is the free variable.*

**Property 13** (Existential Leaves)**.** *For any spanning tree by using the free variable as the root, the leaf node of the query graph can be an existential variable rather than a constant entity.*

**Assumption 14** (Reverse relation enrichment)**.** *For any relation $r$, there exists a reversed $r'$ such that $r'(a, b) = r(b, a)$ for any pair of entities $a, b$.*

**Lemma 15.** *A query that has a sentence as its subformula is trivial and should not be considered.*

We note Assumption 14 is a common practice in previous research (Lacroix et al., 2018), as well as the Lemma 15 (Ren et al., 2020; Ren & Leskovec, 2020).

**Theorem 16.** *With assumption 14 and lemma 15, any $\text{EFO}_1$ query with a simple acyclic query graph that does not have Property 12 and 13 is a TF query.*

In this way, we fill the syntactic gap by making it clear what kinds of queries are left to be discussed, in order to achieve the ambitious goal of answering $\text{EFO}_1$ queries. The proof of Lemma 15 and Theorem 16 is detailed in Appendix A.1.

### 4.3 COMPLEXITY GAP BETWEEN TF AND $\text{EFO}_1$ QUERIES

In this part, we further rethink the complexity of the currently investigated queries, particularly, the traditional claim that "reasoning involves an exponential growth in computational time" (Ren & Leskovec, 2020; Chen et al., 2022).

When discussing the complexity, we assume the knowledge graph $\mathcal{G}$ is complete for simplicity. For general $\text{EFO}_1$ queries, it has long been known to be **NP-complete** (Chandra & Merlin, 1977).

However, TF queries discussed in previous literature are particularly simple. Queries in $\text{TF} \cap \text{EFO}_1$, are well-known special cases with structure-based **tractability** in inference (Vardi, 2000). Specifically, we adapt the complexity results from Dechter & Pearl (1987) on knowledge graphs.

**Proposition 17** (Adapted from Dechter & Pearl (1987))**.** *The optimal complexity of answering TF queries in previous datasets (Ren & Leskovec, 2020; Wang et al., 2021) is $O(nk^2)$, where $n$ is the number of terms and $k$ is a coefficient to characterize the sparsity of the knowledge graph*

$$k = \max_{r \in \mathcal{R}} |\{a \in \mathcal{E} | \exists b.(a, r, b) \in \mathcal{G} \text{ or } (b, r, a) \in \mathcal{G}\}| \tag{4}$$

The more detailed proof is provided in Appendix A.2 where we show we extend traditional results that can only be applied to $\text{TF} \cap \text{EFO}_1$. We found that the complexity grows linearly with the number of variables in the query. The complexity depends on the property of knowledge graphs (see the coefficient $k$) and is at most quadratic. Our theorem provides a strict lower bound for previous optimization methods like QTO (Bai et al., 2023).

We note that the entire $\text{EFO}_1$ queries are inherently harder than TF queries. To the best of our knowledge, none of the existing methods have considered $\text{EFO}_1$-TF queries, and their models lack the ability to represent those queries precisely in syntax.

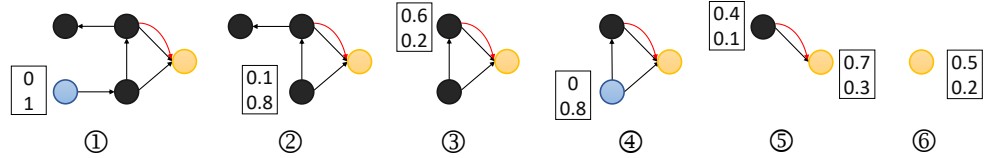

Figure 4: A toy example to show the process of FIT, the vector indicates the fuzzy set of the corresponding node has been updated in this step. The query graph follows Figure 3 with the grounded entity and relation omitted.

## 5 METHODOLOGY

This section presents Fuzzy Inference with Truth values (FIT) algorithm for answering $\text{EFO}_1$ queries on knowledge graphs. FIT is a neural-symbolic method that fine-tunes neural link predictors to address the open-world assumption. Specifically, FIT accepts a wide range of models as long as they provide a "truth value" for a given triple (Trouillon et al., 2016; Cohen, 2016; Zhu et al., 2021). Moreover, we prove that our algorithm ensures faithfulness and perfectness with assumptions.

### 5.1 FUZZY TRUTH VALUE DEFINITION

For simplicity, we follow Cohen (2016) to conceptualize the neural link predictor as $|\mathcal{R}|$ matrices $P_r \in [0,1]^{|\mathcal{E}| \times |\mathcal{E}|}$, where $P_r(a,b)$ is the truth value of triple $(a,r,b)$. We note that $r,a,b$ can be treated as integers by indexing all relations and entities.

It is straightforward to use fuzzy logic theory, and define the truth value function $T$ as the following:

**Definition 18** (Truth value of existential formulas). *Let $\phi$ and $\psi$ be existential formulas, $\top$ and $\bot$ are t-norms and t-conorms, $\bot^\star$ is another t-conorm, and $r \in \mathcal{R}$, $a,b \in \mathcal{E}$.*

*(i)* $T(r(a,b)) = P_r(a,b)$
*(ii)* $T(\neg\phi) = 1 - T(\phi)$
*(iii)* $T(\phi \wedge \psi) = T(\phi)\top T(\psi)$, $T(\phi \vee \psi) = T(\phi)\bot T(\psi)$
*(iv)* $T(\exists x \phi(x)) = \bot^\star_{a \in \mathcal{E}} T(\phi(a))$

For the introduction of the $t$-norm and $t$-conorm, please refer to Appendix C. This definition follows the inductive definition of the existential formula in Definition 3, so if the formula is a sentence, it has a certain truth value. Moreover, we note that $\bot^\star$ is commonly chosen to be Godel $t$-norm (Klir & Yuan, 1995; Hájek, 2013) but also extended to others in recent research (van Krieken et al., 2022).

Then, to answer a $\text{EFO}_1$ query is to estimate the truth values of all possible answers as the corresponding substitutions. The more plausible answers are expected to have a higher truth value. We store the truth values in the answer vector as our goal.

**Definition 19** (Answer Vector). *For $\text{EFO}_1$ query, $\phi(y)$, its answer vector $A[\phi(y)] \in [0,1]^{|\mathcal{E}|}$ is given by $A[\phi(y)](a) = T(\phi(a/y))$.*

### 5.2 DIFFERENTIABLE NEURAL MATRIX CONSTRUCTION

Neural link predictor is a differentiable model that provides a scoring function $s$ for each possible triple (a,r,b). To fit into our definition and construct the neural matrices, we calibrate the real number score $s(a,r,b)$ to probability within $[0,1]$ interval by using the softmax function:

$$P^\star_{r,a}(b) = \frac{exp(s(a,r,b))}{\Sigma_{c \in \mathcal{E}} exp(s(a,r,c))}$$

As the softmax outputs a vector that has the sum of 1, we further compute the scaling:

$$Q_{a,b} = \begin{cases} \frac{|\{d|(a,r,d) \in \mathcal{G}_o\}|}{\Sigma_{c \in \{d|(a,r,d) \in \mathcal{G}_o\}} P^\star_{r,a}(c)}, & \text{if } |\{d|(a,r,d) \in \mathcal{G}_o\}| > 0 \\ 1, & \text{if } |\{d|(a,r,d) \in \mathcal{G}_o\}| = 0 \end{cases} \tag{5}$$

Therefore, the $a$-th row of $r$-th matrix is got by clamping the value for each element:

$$P_r(a,b) = min(1, P^\star_{r,a}(b) \times Q_{a,b}) \tag{6}$$

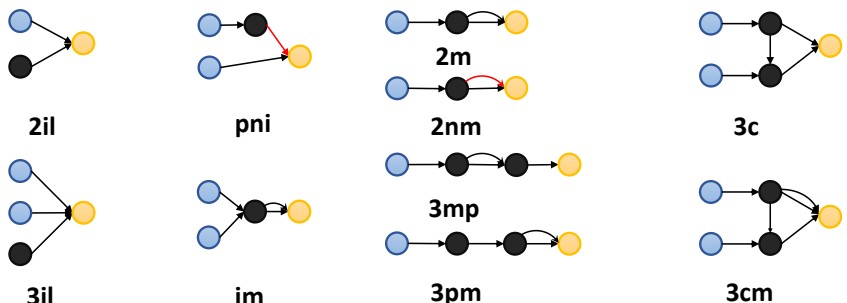

Figure 5: Query graphs of each real $EFO_1$ formula. Naming convention: l "existential leaf", m "multi graph", c for "circle". We also follow the previous convention: "i" for intersection, "p" for projection, and "n" for negation. The representation of query graphs follows Figure 3.

In testing, the construction of the neural matrix is a bit complicated, details in Appendix F.

### 5.3 FORWARD INFERENCE WITH TRUTH VALUE

We explain how to compute the answer vector given neural matrices by a toy example. Full detail of the rigorous derivation is in Appendix D and the analysis of its complexity is in Appendix E. We infer a fuzzy vector $C_u \in [0,1]^{|\mathcal{E}|}$ to represent the fuzzy set for every node $u$ in the query graph, with this help, we remove node and edge step by step and updating $C_u$ for corresponding nodes simultaneously, keeping the answer vector the same on a smaller graph. Our techniques extend traditional symbolic search methods, like Yannakakis (1981) in database theory and Dechter & Pearl (1987) in constrain satisfaction problem, see more discussion in Appendix G.1.

To illustrate our FIT briefly, we explain each step shown in Figure 4.3. 1. We initialize the constant entity by giving it a one-hot vector as its fuzzy set. 2. We remove the constant entity and update the fuzzy set of the nodes that are connected to the constant. 3. We remove a leaf variable and update the fuzzy vector of its neighbor. 4. Since it is a circle in the graph, we enumerate possible candidates in the fuzzy set of a variable and make it a constant. 5. Similar to step 2, we remove the constant. 6. The query graph contains only the free variable and the final answer is naturally its fuzzy set. We note this example has included all three features that should be considered according to our Theorem 16: existential leaf, multigraph, and cyclic graph.

Finally, as all the computations in the FIT are differentiable, the loss is defined as cross-entropy of the predicted answer vector $A$ with the real answer $\mathcal{A}$:

$$\mathcal{L} = H(A, \mathcal{A}) = -[\Sigma_{a \in \mathcal{A}} \ln(A(a)) + \Sigma_{(a \in \mathcal{E} - \mathcal{A})} \ln(1 - A(a))] \tag{7}$$

In this way, backward propagation helps to fine-tune the neural link predictors using data of the complex query, rather than just one-hop queries.

### 5.4 THEORETICAL GUARANTEES

**Definition 20** (Perfect Matrices). *Given a knowledge graph $\mathcal{G}$, we say the matrices $\{P_r\}_{r \in \mathcal{R}}$ are perfect if $P_r(h,t) = 1 \iff (h,r,t) \in \mathcal{G}$, $P_r(h,t) = 0 \iff (h,r,t) \notin \mathcal{G}$.*

**Theorem 21** (Perfectness). *If the matrices $\{P_r\}_{r \in \mathcal{R}}$ are perfect, then the FIT algorithm is perfect, meaning: $A[\phi(y)](a) = 1 \iff a \in \mathcal{A}[\phi(y)]$, $A[\phi(y)](a) = 0 \iff a \notin \mathcal{A}[\phi(y)]$.*

Though there is no perfect model in practice, we can always assume the given model is "good", which admits the following consistency assumption.

**Assumption 22** (Consistent matrix). *For any observed triple $(a,r,b) \in \mathcal{G}_o$, $P_r(a,b) = 1$, for any unobserved triple $(a,r,b) \notin \mathcal{G}_o$, $P_r(a,b) < 1$.*

**Definition 23.** *We say a model is **faithful** if it is able to retrieve **deductible** answers like a traditional logical inference system like database query (Kroenke, 2002; Sun et al., 2020).*

**Theorem 24** (Faithfulness). *With assumption 22, for any $EFO_1$ query $\phi(y)$ without negation, FIT reaches perfect faithfulness, meaning that every answer $a$ that can be deduced in the observed knowledge graph $\mathcal{G}_o$, $A[\phi(y)](a) = 1$.*

Table 1: MRR results(%) of the new queries on the real $EFO_1$ dataset.

| Knowledge Graph | Method | pni | 2il | 3il | 2m | 2nm | 3mp | 3pm | im | 3c | 3cm | AVG. |
|---|---|---|---|---|---|---|---|---|---|---|---|---|
| FB15k-237 | BetaE | 9.0 | 25.0 | 40.1 | 8.6 | 6.7 | 8.6 | 6.8 | 12.3 | 25.2 | 22.9 | 16.5 |
| | LogicE | 9.5 | 27.1 | 42.0 | 8.6 | 6.7 | 9.4 | 6.1 | 12.8 | 25.4 | 23.3 | 17.1 |
| | ConE | 10.8 | 27.6 | 43.9 | 9.6 | 7.0 | 9.3 | 7.3 | 14.0 | 28.2 | 24.9 | 18.3 |
| | QTO | 12.1 | 28.9 | 47.9 | 8.5 | 10.7 | 11.4 | 6.5 | 17.9 | 38.3 | 35.4 | 21.8 |
| | CQD | 7.7 | 29.6 | 46.1 | 6.0 | 1.7 | 6.8 | 3.3 | 12.3 | 25.9 | 23.8 | 16.3 |
| | LMPNN | 10.7 | 28.7 | 42.1 | 9.4 | 4.2 | 9.8 | 7.2 | 15.4 | 25.3 | 22.2 | 17.5 |
| | FIT | **14.9** | **34.2** | **51.4** | **9.9** | **12.7** | **11.9** | **7.7** | **19.6** | **39.4** | **37.3** | **23.9** |
| FB15k | BetaE | 29.9 | 34.8 | 50.6 | 24.4 | 9.6 | 19.0 | 18.4 | 29.1 | 30.5 | 30.7 | 27.7 |
| | LogicE | 30.7 | 39.3 | 53.0 | 24.1 | 10.5 | 20.5 | 15.5 | 30.7 | 31.8 | 31.7 | 28.8 |
| | ConE | 37.0 | 40.1 | 57.3 | 33.3 | 11.5 | 23.9 | 27.6 | 38.7 | 35.0 | 36.3 | 34.1 |
| | QTO | 48.2 | 49.5 | 68.2 | 64.6 | 19.4 | 48.5 | 53.7 | 73.9 | 53.3 | 54.9 | 53.4 |
| | CQD | 24.2 | 47.6 | 65.4 | 23.2 | 1.6 | 11.0 | 8.7 | 36.3 | 31.3 | 32.9 | 28.2 |
| | LMPNN | 38.7 | 43.2 | 57.8 | 40.3 | 7.9 | 24.0 | 30.5 | 48.4 | 32.2 | 30.9 | 35.4 |
| | FIT | **57.9** | **70.4** | **77.6** | **73.5** | **39.1** | **57.3** | **64.0** | **79.4** | **63.8** | **65.4** | **64.8** |
| NELL | BetaE | 7.5 | 43.3 | 64.6 | 29.0 | 5.3 | 8.7 | 14.4 | 29.5 | 36.1 | 33.7 | 27.2 |
| | LogicE | 9.8 | 47.0 | 66.6 | 34.7 | 6.4 | 13.3 | 17.8 | 35.1 | 38.9 | 37.9 | 30.8 |
| | ConE | 10.3 | 42.1 | 65.8 | 32.4 | 7.0 | 12.6 | 16.8 | 34.4 | 40.2 | 38.2 | 30.0 |
| | QTO | 12.3 | 48.5 | 68.2 | 38.8 | 12.3 | 22.8 | 19.3 | 41.1 | 45.4 | 43.9 | 35.3 |
| | CQD | 7.9 | 48.7 | 68.0 | 31.7 | 1.5 | 12.9 | 13.8 | 33.9 | 38.8 | 35.9 | 29.3 |
| | LMPNN | 11.6 | 43.9 | 62.3 | 35.6 | 6.2 | 15.9 | 19.3 | 38.3 | 39.1 | 34.4 | 30.7 |
| | FIT | **14.4** | **53.3** | **69.5** | **42.1** | **12.5** | **24.0** | **22.8** | **41.5** | **47.5** | **45.3** | **37.3** |

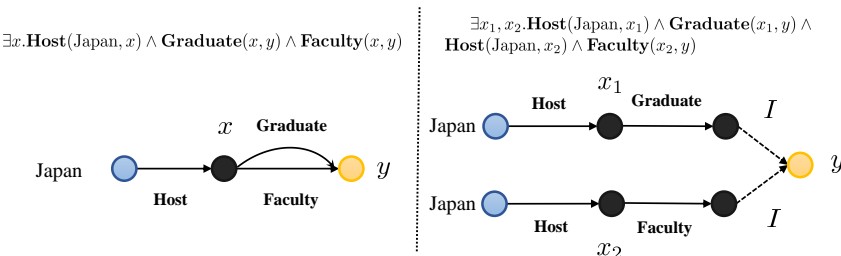

Figure 6: Left: The $EFO_1$ version. Right: The Tree-Form approximation. The original $x$ is split into $x_1, x_2$ in the Tree-Form query, leading to a different semantic meaning. The formal first order logic formula is also given at the top.

The proof of the Theorem 21 and Theorem 24 is provided in Appendix A.3.

## 6  REAL $EFO_1$ DATASET

Based on our previous discussion of the limitation of query embedding methods in Section 4.2, we have crafted ten real $EFO_1$ formulas: pni, 2il, 3il, im, 2m, 2nm, 3mp, 3pm, 3c, 3cm. We note these formulas have included all possible new properties of the $EFO_1$ formulas: Property 12(pni), Property 13(2il,3il), multi-graph(im, 2m, 2nm, 3mp, 3pm, 3cm), and cyclic graph(3c, 3cm). Graphs with self-loop are not included, explained in Appendix I. Additionally, the formula "pni" has already been put forward by previous dataset (Ren & Leskovec, 2020), but it is answered as the universal quantifier version as illustrated in Example 10, we maintain the query but re-sample its answer according to our definition. The query graphs of our new formulas are presented in Figure 5. Its statistics are given in Appendix J.

## 7  EXPERIMENT

### 7.1  SETTINGS

We evaluate our algorithm on various tasks. Firstly, we evaluate our algorithm on our new dataset of real $EFO_1$ queries developed in Section 6 and show the failure of existing methods. Secondly, we compare our algorithm with existing methods on the dataset of Tree-Form queries provided by Ren & Leskovec (2020). Thirdly, we verify our claims of Theorem 24 by evaluating also on Tree-Form.

We follow the standard protocol, splitting the answer into two parts: **deductible** answers which can be found by the observed knowledge graph $\mathcal{G}_o$, corresponds to Section 7.4, **predicted** answers that need generalization which can be found in the whole knowledge graph $\mathcal{G}$ but not from $\mathcal{G}_o$, correspond to Section 7.2 and 7.3. All our experiments use Mean Reciprocal Rank (MRR) as the metric.

Table 2: MRR results(%) of the Tree-Form queries. The average score is computed separately among positive and negative queries. The scores of CQD and LMPNN is directly taken from their paper (Minervini et al., 2022; Wang et al., 2023b).

| KG | Method | 1p | 2p | 3p | 2i | 3i | ip | pi | 2u | up | AVG.(P) | 2in | 3in | inp | pin | AVG.(N) |
|---|---|---|---|---|---|---|---|---|---|---|---|---|---|---|---|---|
| | CQD | **46.7** | 13.3 | 7.9 | 34.9 | 48.6 | 20.4 | 27.1 | 17.6 | 11.5 | 25.3 | | | | | |
| FB15k-237 | LMPNN | 45.9 | 13.1 | 10.3 | 34.8 | 48.9 | 17.6 | 22.7 | 13.5 | 10.3 | 24.1 | 8.7 | 12.9 | 7.7 | 4.6 | 8.5 |
| | FIT | **46.7** | **14.6** | **12.8** | **37.5** | **51.6** | **21.9** | **30.1** | **18.0** | **13.1** | **27.4** | **14.0** | **20.0** | **10.2** | **9.5** | **13.4** |
| | CQD | 89.2 | 65.3 | 29.7 | 77.4 | 80.6 | 71.6 | 70.6 | 72.3 | **59.4** | 68.5 | | | | | |
| FB15k | LMPNN | 85.0 | 39.3 | 28.6 | 68.2 | 76.5 | 43.0 | 46.7 | 36.7 | 31.4 | 50.6 | 29.1 | 29.4 | 14.9 | 10.2 | 20.9 |
| | FIT | **89.4** | **65.6** | **56.9** | **79.1** | **83.5** | **71.8** | **73.1** | **73.9** | 59.0 | **72.5** | **40.2** | **38.9** | **34.8** | **28.1** | **35.5** |
| | CQD | 60.4 | 22.6 | 13.6 | 43.6 | 53.0 | 25.6 | 31.2 | 19.9 | 16.7 | 31.8 | | | | | |
| NELL | LMPNN | 60.6 | 22.1 | 17.5 | 40.1 | 50.3 | 24.9 | 28.4 | 17.2 | 15.7 | 30.8 | 8.5 | 10.8 | 12.2 | 3.9 | 8.9 |
| | FIT | **60.8** | **23.8** | **21.2** | **44.3** | **54.1** | **26.6** | **31.7** | **20.3** | **17.6** | **33.4** | **12.6** | **16.4** | **15.3** | **8.3** | **13.2** |

Table 3: MRR results(%) of the deductible answers in the BetaE dataset.

| Formula | 1p | 2p | 3p | 2i | 3i | pi | ip | 2in | 3in | inp | pin | pni | 2u | up |
|---|---|---|---|---|---|---|---|---|---|---|---|---|---|---|
| MRR | 100 | 100 | 100 | 100 | 100 | 100 | 100 | 75.5 | 65.3 | 65.2 | 65.7 | 90.5 | 100 | 100 |

The discussion of the choice of hyperparameters, including the $t$-norm/conorm, is in Appendix H.

## 7.2 EXPERIMENT ON REAL EFO$_1$ DATASET

Here we present our result on our new proposed dataset, shown in Table 1. As explained in Section 3, the Tree-Form query fails to represent the same syntax as our newly developed real EFO$_1$ queries so it can only **syntactically approximate** the desired real EFO$_1$ query. We offer a detailed example of our new "2m" query in Figure 6, where the semantics of the left is "Find someone who is a graduate of a Japanese University and also a faculty of the **same** university" while the right means "Find someone who is a graduate of a Japanese University and is also a faculty of a Japanese university". The apparent difference illustrated by this example explains why the previous method falls far behind our FIT, in all types of query and all knowledge graphs. For more details on the implementation of the baseline methods, please refer to Appendix K.

## 7.3 EXPERIMENTS ON EXISTING BETAE DATASET (TREE-FORM)

As our FIT utilizes a pretrained neural link predictor, for fairness, we use the same checkpoint provided by CQD (Minervini et al., 2022) to compare. We note that LMPNN (Wang et al., 2023b) also builds its model by the same checkpoint and adds its own new learnable parameters, thus LMPNN is also added as a baseline. Additionally, the connection to QTO is explained in Appendix G.2, where we show how FIT is a pure extension to QTO.

Since CQD is not able to answer queries with negation, we split the experiment result into the positive part and the negative part. The result of the positive queries is shown in Table 2. We have found that our FIT algorithm surpasses both baselines in all knowledge graphs. Most significantly, when the diameter of the query graph gets bigger, like 3p queries, the advantage of our FIT are enormous. As for the result of negative queries, FIT outperforms LMPNN by a large margin in all query types.

## 7.4 EXPERIMENTS ON FAITHFULNESS

To verify our Theorem 24, we evaluate our algorithm by the **deductible** answers. The result is shown in Table 3. To the best of our knowledge, Sun et al. (2020) is the only one that considers faithfulness, we omit its result since it can only infer positive queries in which we have reached perfect faithfulness.

## 8 CONCLUSION

We reviewed the limitations of current query embedding methods and extended the scope of complex query answering to the whole family of EFO$_1$ formulas. We also developed a new dataset containing ten formulas and analyzed the new difficulties coming with these formulas. Finally, based on strict fuzzy logic theory, we present a new neural-symbolic method, FIT, which can fine-tune pretrained neural link predictor to infer arbitrary EFO$_1$ formula and outperform existing methods significantly.

ACKNOWLEDGMENTS

The authors of this paper were supported by the NSFC Fund (U20B2053) from the NSFC of China, the RIF (R6020-19 and R6021-20) and the GRF (16211520 and 16205322) from RGC of Hong Kong. We also thank the support from the UGC Research Matching Grants (RMGS20EG01-D, RMGS20CR11, RMGS20CR12, RMGS20EG19, RMGS20EG21, RMGS23CR05, RMGS23EG08).

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

## A MISSING PROOFS IN THE MAIN PAPER

We offer all the proof of the results in the main paper in this section.

### A.1 PROOF OF LEMMA 15 AND THEOREM 16

Firstly we proof the lemma 15 by stating it very clearly as the following:

**Lemma 25.** *A query that has a sentence as its subformula is trivial and should not be considered. To be specific, suppose the formula reads $\phi(y_1, \cdots, y_n)$, then it falls into one of the two situations: 1. $\phi(y_1, \cdots, y_n)$ has a certain truth value of 1 or 0, irrelevant of the choice of $y_1, \cdots, y_n$, therefore trivial. 2. There is a subformula $\psi(y_1, \cdots, y_n)$ of the original $\phi(y_1, \cdots, y_n)$ that has the same answer set, meaning the original $\phi(y_1, \cdots, y_n)$ is redundant and should not be considered.*

*Proof.* We prove that recursively.

Given a sentence we call $\chi$, w.r.o.t, we assume its truth value is 1.

- For another query $\psi$, $\psi \wedge \chi$ has the same answer set with $\psi$, which falls into the second situation.

- For another query $\psi$, $\psi \vee \chi$ always has truth value 1, which falls into the first situation.

- $\neg\chi$ has a certain truth value of 0, which falls into the first situation.

- $\exists x \chi$ and $\forall x \chi$ both have a certain truth value of 1 (assume the knowledge graph is not empty), which falls into the first situation.

By the recursive definition of first order logic formulas, this lemma is proved. □

We apply this lemma to acyclic queries:

**Lemma 26.** *With lemma 15, if a* $\text{EFO}_1$ *query* $\phi(y)$ *has an acyclic query graph, then each of its constant entities should be a leaf node in the query graph.*

*Proof.* We prove by contradiction: if there exists node $a$ which is a non-leaf node but it is also a constant entity. Then we consider the sub-tree whose root is $a$, which represents an existential formula $\psi$ without a free variable (recall that the only free variable $y$ is the root of the whole tree), thus it is a sentence. By lemma 15, we exclude this kind of query. □

Then we give the proof of Theorem 16:

*Proof.* Considering query $\phi(y)$ whose query graph is acyclic, we compute the spanning tree of it by using the free variable $y$ as the root. Assumption 14 ensures that each edge is from the child node to its parent node. With this representation, we prove the theorem by mathematical induction on the number of nodes in the query graph.

We start when there are two nodes in the query graph as one node can not construct an edge. Then the query must have a form of $r(a, y)$ or $\neg r(a, y)$ by it does not have property 13. Either way, it is a TF query by definition 8.

Assuming this holds for query graphs with no more than $n \geq 2$ nodes, then we prove it for graphs with $n + 1$ nodes.

Find the root $y$, using lemma 26, we know it only connects to existential variables or leaf nodes.

If $y$ connects to a leaf node $u$, we know $u$ must be an entity $a$ by not having property 13. Then $\phi(y) = r(a, y) \wedge \psi(y)$, where $\psi(y)$ is an TF query by induction. Then using the third part of definition 8 finishes the proof.

If $y$ only connects to existential variables, find one child node $u$ with the edge $r$, we note the edge must be positive because of not having property 12. Then $\phi(y)$ can be decomposed as $\phi(y) = r(u, y) \wedge \psi(u) \wedge \chi(y)$, where $\psi(u)$ and $\chi(y)$ are both TF query by induction (If $y$ only connected to $u$, $\chi(y)$ is omitted). Then using the third and fourth part of definition 8 finishes the proof.

Then, the proof is finished by induction. □

## A.2    PROOF OF THE COMPLEXITY OF TREE FORM QUERY

To begin with, we note that in the construction of the current dataset, there is another assumption called "bounded negation" that has been proposed by Ren & Leskovec (2020) and inherited by Wang et al. (2021), which makes sure that set complement is "bounded" by set intersection:

**Proposition 27** (Bounded Negation). *In every operator tree of the previous datasets (Ren & Leskovec, 2020; Wang et al., 2021), every node representing "N" must have a parent node "I", which must have another child node that does not represent "N".*

Then, we follow the algorithm proposed in Dechter & Pearl (1987), after sorting the query as the operator tree in the DNF, which is just $O(n)$ time, as there are $n$ variables in the operator tree, there are only $n - 1$ projection edges in the operator tree. We just need to prove that in computing the answer set for every variable in the operator tree, each projection edge has the complexity of $O(k^2)$. Specifically, we prove a different lemma using the mathematical induction:

**Lemma 28.** *For every variable node $u$ in the operator tree, as we have pointed out in Appendix B, it represents the answer set of a sub Tree-Form query $\phi(u)$, we prove its answer set $A[\phi(u)] \leq k$.*

*Proof.* For the leaf variable, since they are all constants, each one just needs $O(1)$ time to start with.

For a variable $v$, assuming all its child nodes in the operator tree have been computed. By the Proposition 27, we know that there is always a child node $u$ such that there is a positive edge $r$ linked from $u$ to $v$ in the operator tree.

We check the set projection from $u$ to $v$ has the complexity of $O(k^2)$:

Given the answer set $U$ corresponding to node $u$ and satisfying that $|U| \leq k$, given the relation $r$, we prove that the set projection of $U$ satisfies that $|\{v|\exists u \in U. \textbf{ s.t. } r(c, b) \in \mathcal{G}\}| \leq k$, this is immediately right because of the definition of $k$.

Then by the nature of the set intersection, the size of the answer set of node $v$ is at most $k$(set union is omitted since the query is in DNF). In this way, we finish the proof by induction. $\square$

Then, we restate the original proposition here and prove it:

**Proposition 29** (Adapted from Dechter & Pearl (1987)). *The optimal complexity of answering* TF *queries in previous datasets (Ren & Leskovec, 2020; Wang et al., 2021) is $O(nk^2)$, where $n$ is the number of terms and $k$ is a coefficient defined by the knowledge graph*

$$k = \max_{r \in \mathcal{R}} |\{a \in \mathcal{E}|\exists b.(a, r, b) \in \mathcal{G} \text{ or } (b, r, a) \in \mathcal{G}\}| \tag{8}$$

*Proof.* By the lemma above, we show that every projection edge is answered in $O(k^2)$ time. Since set complement, intersection, and union only takes $O(k)$ time and the number of these operations is also $O(n)$, we finish the proof that solving the whole Tree-Form query in the Constraint Satisfaction Problem way has the complexity of $O(nk^2)$. $\square$

### A.3 PROOF OF THE THEORETICAL GUARANTEES OF FIT

In this part, we give proof of the theoretical guarantees of FIT.

**Theorem 30** (Perfectness). *If the matrices$\{P_r\}_{r \in \mathcal{R}}$ are perfect, then the FIT algorithm is perfect, meaning: $A[\phi(y)](a) = 1 \iff a \in \mathcal{A}[\phi(y)]$, $A[\phi(y)](a) = 0 \iff a \notin \mathcal{A}[\phi(y)]$.*

*Proof.* Our definition of fuzzy truth value coincides with conventional truth value when the truth value of any atomic formula is in $\{0, 1\}$ (Thiele, 1994). Thus, if the given matrices are perfect, our computed truth value coincides with the conventional truth value defined in first order logic. $\square$

**Theorem 31** (Faithfulness). *With assumption 22, for any $EFO_1$ query $\phi(y)$ without negation, FIT reaches perfect faithfulness, meaning that every answer $a$ that can be deduced in the observed knowledge graph $\mathcal{G}_o$, $A[\phi(y)](a) = 1$.*

*Proof.* We create perfect matrices by the observed knowledge graph $\mathcal{G}_o$ as in Definition 20, the truth value function induced by it is $T'$. By Theorem 21, answer $a$ is deductible in $\mathcal{G}_o$ if and only if $T'(a) = 1$. We note that $T(r(b, c)) \geq T'(r(b, c))$ for any $r \in \mathcal{R}, b, c \in \mathcal{E}$, then by the non-decreasing of $t$-norm and $t$-conorm, we know that $T(\phi(a)) \geq T'(\phi(a)) = 1$. Thus $A[\phi(y)](a) = T(\phi(a)) = 1$. $\square$

# B    TREE FORM QUERY DERIVATION

In this section, we explain the derivation of the tree form query which we introduce in Definition 8.

Given a relation $r$, the set projection of set $B$ is $\{c|\exists b \in B. \textbf{ s.t. } r(b, c) \in \mathcal{G}\}$.

For $\phi(y) = r(a, y)$, $\mathcal{A}[\phi(y)] = \{b|r(a, b) \in \mathcal{G}\}$, which is projection of a single entity.

For general projection,

$$a \in \mathcal{A}[\exists y.r(y, y') \wedge \phi(y)]$$
$$\Longleftrightarrow \exists b \in \mathcal{E}.(T(r(b, a)) = 1) \wedge (T(\phi(b)) = 1)$$
$$\Longleftrightarrow \exists b \in \mathcal{E}.r(b, a) \in \mathcal{G} \wedge b \in \mathcal{A}[\phi(y)]$$

Thus we know that $\mathcal{A}[\exists y.r(y, y') \wedge \phi(y)]$ is projection of set $\mathcal{A}[\phi(y)]$ with relation $r$.

The derivation of negation:

$$a \in \mathcal{A}[\neg\phi(y)]$$
$$\Longleftrightarrow T(\neg\phi(a/y)) = 1$$
$$\Longleftrightarrow T(\phi(a/y)) = 0$$
$$\Longleftrightarrow a \notin \mathcal{A}[\phi(y)]$$
$$\Longleftrightarrow a \in \overline{\mathcal{A}[\phi(y)]}$$

where $\overline{\mathcal{A}[\phi(y)]}$ represents the complement set of $\mathcal{A}[\phi(y)]$. And we know $\mathcal{A}[\neg\phi(y)] = \overline{\mathcal{A}[\phi(y)]}$.

The derivation of conjunction is given as follows:

$$a \in \mathcal{A}[(\phi \wedge \psi)(y)]$$
$$\Longleftrightarrow T((\phi \wedge \psi)(a)) = 1$$
$$\Longleftrightarrow (T(\phi(a)) = 1) \wedge (T(\psi(a)) = 1)$$
$$\Longleftrightarrow a \in \mathcal{A}[\phi(y)] \wedge a \in \mathcal{A}[\psi(y)]$$

Thus we know $\mathcal{A}[(\phi \wedge \psi)(y)] = \mathcal{A}[\phi(y)] \cap \mathcal{A}[\psi(y)]$. The derivation of disjunction is the same.

# C    $t$-NORM INTRODUCTION

**Definition 32** ($t$-norm). *A $t$-norm $\top$ is a function: [0,1] x [0,1] $\rightarrow$ [0,1] that satisfies the following properties:*

*(i) Communitavity: $a\top b = b\top a$*

*(ii) Monotonicity: $(a\top b) \leq (c\top b)$ if $a \leq c$*

*(iii) Associativity: $(a\top b)\top c = a\top(b\top c)$*

*(iv) Neutrality: $a\top 1 = a$*

Then the $t$-conorm $\perp$ is directly defined by $a\perp b = 1 - (1 - a)\top(1 - b)$, which follows the De Morgan's law.

Finally, we introduce some common $t$-norms which are of interest:

(i) Godel: $a\top_G b = \min(a, b)$
(ii) Product: $a\top_P b = a * b$
(iii) Łukasiewicz: $a\top_{LK} b = \max(a + b - 1, 0)$

In the main paper, we mainly focus on the Godel and Product $t$-norm.

## D  FIT METHODOLOGY DETAILS

In this section, we give the details of the proof used in Section 5. We elaborate the Fuzzy Inference with Truth Value (FIT) algorithm by the direct derivation from our previous truth value definition. We discuss how to derive the answer vector of conjunctive queries $\gamma_j(y)$. The answer vector of the whole formula can then be computed by Definition 18. The main idea is to cut node and edge step by step and infer on a smaller and smaller query graph.

To begin with, we define a new membership predicate (denoted as $\mu$) between a variable $u$ and a vector $C_u$ representing its fuzzy set, which leads to the truth value of whether entity $a$ belongs to a fuzzy set.

**Definition 33.** *Given an entity $a$ and a vector $C \in [0,1]^{|\mathcal{E}|}$, the truth value of membership predicate $\mu$ is $T(\mu(a,C)) = C(a)$, namely, the $a$-th coordinate of the vector $C$.*

Equipped with those definitions, we are allowed to build our algorithm on a solid fuzzy logic background (Klir & Yuan, 1995).

### D.1  STEP 1: INITIALIZATION

We initialize the fuzzy vectors for every non-entity node in the query graph $G(\gamma_j)$: for node $u$ indicates either an existential variable or the free variable, we initialize $C_u = \mathbf{1}$ by an all-one fuzzy vector. The original query is updated as $\gamma(y) \wedge \mu(u, C_u)$.

Let the derived query be $\phi(y)$, which is $\gamma(y)$ conjuncted with membership predicates from all non-entity nodes.

### D.2  STEP 2: REMOVE SELF-LOOPS

For any node $u$ that has self-loop, we discuss two situations, (1) $u$ is the answer node, and (2) $u$ is not the answer node. We note that the self-loop of a constant entity makes no sense.

**Case1:Node $u = y$ is the answer node.** In this case, we randomly select one of its self-loop edges, if it is positive, representing $r(y,y)$, putting every formula that contains $y$ forehead, the query formula $\phi(y)$ reads

$$\phi(y) = \mu(y, C_y) \wedge r(y,y) \wedge \psi(y)$$

where $\psi(y)$ is a sub formula.

The answer vector $A[\phi(y)]$ is inferred by evaluating the truth value $T(\phi(a/y))$, then for all $a \in \mathcal{E}$,

$$\begin{aligned} A[\phi(y)](a) = T(\phi(a)) &= T(\mu(y, C_y) \wedge r(y,y) \wedge \psi(y)) \\ &= C_y(a) \top P_r(a,a) \top T(\psi(a)) \end{aligned}$$

By introducing $\odot^\top$ as applying $t$-norm $\top$ element-wise, we have the following vector form:

$$\begin{aligned} A[\phi(y)] &= (C_y \odot^\top \text{diag}(P_r)) \odot^\top A[(\psi(y))] \\ &= \mu(y, C_y \odot^\top \text{diag}(P_r)) \wedge \psi(y) \end{aligned}$$

By this, we show that the self-loop edge can be removed as long as we update the corresponding fuzzy vector simultaneously.

If the edge is negative, meaning it represents a formula $\neg r(y,y)$, the derivation is similar:

$$\begin{aligned} A[\phi(y)](a) = T(\phi(a)) &= T(\mu(a, C_y) \wedge \neg r(a,a) \wedge \psi(a)) \\ &= C_y(a) \top (1 - P_r(a,a)) \top T(\psi(a)) \\ A[\phi(y)] &= A[\mu(y, C_y \odot^\top (1 - \text{diag}(P_r))) \wedge \psi(y)] \end{aligned}$$

Thus, $A[\phi(y)]$ can be derived once we infer the answer vector of $A[\psi(y)]$, where $\psi(y)$ is the simplified, inner formula.

**Case 2:If $u$ represents an existential variable $x$.** Without loss of generality, we assume there is only $n$ positive self-loops $r_1(x,x), \cdots, r_n(x,x)$. Then, the formula reads $\phi(y) = \exists x. \mu(x, C_x) \wedge r_1(x,x) \wedge \cdots \wedge r_n(x,x) \wedge \psi(y;x)$, where $\psi(y;x)$ is an existential formula with the free variable $y$ and existential variable $x$ which does not have sekf-loop anymore. For $a \in \mathcal{E}$, we have:

$$
\begin{aligned}
&T(\phi(a)) \\
&= \perp_{b \in \mathcal{E}}^{\star}[T(\mu(b, C_x) \wedge r_1(b,b) \wedge \cdots \wedge r_n(b,b) \wedge \psi(a;b)] \\
&= \perp_{b \in \mathcal{E}}^{\star}[C_x(b) \top P_{r_1}(b,b) \top \cdots \top P_{r_n}(b,b) \top T(\psi(a;b))] \\
&= \perp_{b \in \mathcal{E}}^{\star}[C_x'(b) \top T(\psi(a;b))] \\
&= T(\exists x. \mu(x, C_x') \wedge \psi(a/y; x))
\end{aligned}
$$

where $C_x' = C_x \odot^{\top} \operatorname{diag}(P_{r_1}) \odot^{\top} \cdots \odot^{\top} \operatorname{diag}(P_{r_n})$. Therefore, we can remove multiple self-loops similarly.

By the end of this step, all self-loops are removed.

### D.3    Step 3: Remove constant entity

If there is a node that represents an entity $a$, we can cut these edges from $a$ to other nodes easily.

Considering all nodes that are connected to $a$, there are two situations:(1)$a$ connects to the free variable $y$, and (2) $a$ connects to an existential variable $x$.

**Case 1: The edge connects $a$ to the free variable $y$**

As the scenario of negative edge and multi-edge has been discussed, without loss of generality, we can assume there is only one edge from $a$ to $y$ and one edge from $y$ to $a$, which are both positive. We note that we get rid of the Assumption 14 naturally. The query formula $\phi(y)$ reads

$$
\phi(y) = \mu(y, C_y) \wedge r_1(a, y) \wedge r_2(y, a) \wedge \psi(y)
$$

where $\psi(y)$ is a sub formula.

$$
\begin{aligned}
A[\phi(y)](b) &= T(\mu(b, C_y) \wedge r_1(a, b) \wedge r_2(b, a) \wedge \psi(b)) \\
&= C_y(b) \top P_{r_1}(a, b) \top P_{r_2}^{\mathsf{T}}(a, b) \top T(\psi(b)) \\
&= C_y'(b) \top T(\psi(b)) = A[C_y'(y) \wedge \psi(y)](b)
\end{aligned}
$$

where $C_y' = C_y \odot^{\top} P_{r_1}(a) \odot^{\top} P_{r_2}^{\mathsf{T}}(a)$. We also show that the inverse relations can be naturally tackled by the transpose of the predicate matrix.

**Case 2:The edge connects $a$ to an existential variable $x$.**

W.r.o.t, we can assume there is only one edge from $a$ to $x$, the derivation is similar with the one before:

$$
\begin{aligned}
\phi(y) &= \mu(x, C_x) \wedge r(a, x) \wedge \psi(y; x) \\
A[\phi(y)](b) &= \perp_{x \in \mathcal{E}}^{\star}[T(\mu(x, C_x) \wedge r(a, b) \wedge \psi(b; c))] \\
&= \perp_{c \in \mathcal{E}}^{\star}[C_x(c) \top P_r(a, c) \top T(\psi(b; c))] \\
&= \perp_{c \in \mathcal{E}}^{\star}[C_x'(c) \top T(\psi(b; c))] \\
&= A[\mu(x, C_x' \wedge \psi(y))](b)
\end{aligned}
$$

In this way, all edges connected with $a$ is removed, and then node $a$ is also removed. By the end of this step, all nodes in the query graph represents constants are removed as well as these corresponding edges.

### D.4    Step 4: Cutting leaf node

In a query graph, we say a node $u$ is a leaf if it only connects to one other node $v$. If there is a leaf node $u$ in the query graph, We show that we are able to efficiently cut the leaf node and do the inference on the remaining graph, w.r.o.t, we can assume there is only one positive edge from $u$ to $v$.

There are three cases in this step:

**Case 1:$u$ and $v$ represent two existential variables.**

Similarly, w.l.o.g., we can assume the formula reads:

$$\phi(y) = \exists x_1, x_2.\mu(x_1, C_{x_1}) \wedge r(x_1, x_2) \wedge \mu(x_2, C_{x_2}) \wedge \psi(x_2, y)$$

We want to update the candidate vector $C_{x_2}$ instead of really enumerating all possible assignments of $x_1$. That is to say, we want to find $C'_{x_2}$ such that for every possible $a \in \mathcal{E}$, it satisfies:

$$T(\phi(a)) = T(\exists x_2.\mu(x_2, C'_{x_2}) \wedge \psi(x_2, a))$$

For simplification, we define an operation $\circ^\top$ that for matrix $M, N$ and vector $v$, $M \circ^\top v = N \iff N(i,j) = M(i,j)\top V(j)$, let $C^\star = [(P_r \circ^\top C_{x_2})^\intercal \circ^\top C_{x_1}]^\intercal$, and use the commutativity of $t$-conorm, the original formula can be reformulated as:

$$
\begin{aligned}
T(\phi(a)) &= \bot^\star_{x_1=b, x_2=c}[C_{x_1}(b)\top P_r(b,c)\top C_{x_2}(c)\top T(\psi(c,a))] \\
&= \bot^\star_{x_2=c}\{\bot^\star_{x_1=b}[C^\star(b,c)\top T(\psi(c,a))]\}
\end{aligned}
\tag{9}
$$

And the desired form is

$$T(\exists x_2.\mu(x_2, C'_{x_2}) \wedge \psi(x_2, a)) = \bot^\star_{x_2=c}[C'_{x_2}(c)\top T(\psi(c,a))] \tag{10}$$

Compare equation 9 and 10, since $T(\psi(c,a))$ is completely unknown, we want it hold for every $a, c$ that:

$$\bot^\star_{x_1=b}[C^\star(b,c)\top T(\psi(c,a))] = C'_{x_2}(c)\top T(\psi(c,a))$$

We note this can not be done by arbitrary $t$-conorm. However, if the $\bot^\star$ is Godel, namely max, then by the nondecreasing of $t$-conorm we have an important conclusion:

$$\max_b[C^\star(b,c)\top T(\psi(c,a))] = \max_b[C^\star(b,c)]\top T(\psi(c,a)) \tag{11}$$

Then we finally get desired $C'_{x_2}(c) = \max_b[C^\star(b,c)]$.

**Case 2:$u$ is the free variable $y$, $v$ is existential variable $x$.**

Then, the formula $\phi(y)$ reads:

$$\phi(y) = \mu(y, C_y) \wedge [\exists x.r(x,y) \wedge \psi(x)]$$

where $\psi(x)$ is a formula with $x$ being the only free variable.

$$
\begin{aligned}
A[\phi(y)](a) &= C_y(a)\top T(\exists x.r(x,a) \wedge \psi(x)) \\
&= C_y(a)\top\{\bot^\star_{x=b}[P_r(b,a)\top\psi(b)]\}
\end{aligned}
$$

If we define

$$A^\star = P_r \circ^\top A[\psi(x)]$$

where $\circ^\top$ is defined samely as in main paper. We can have the simplification:

$$A[\phi(y)](a) = C_y(a)\top[\bot^\star_{x=b}A^\star(b,a)]$$

In this way, we can derive the final answer vector $A[\phi(y)]$ by computing the answer vector $A[\psi(x)]$ first.

**Case 3:If $u$ is an existential variable $x$ and $v$ is the free variable $y$.** We also assume there is only one edge from $u$ to $v$:

The formula $\phi(y)$ reads:

$$\phi(y) = \exists x.\mu(x, C_x) \wedge r(x,y) \wedge \mu(y, C_y) \wedge \psi(y)$$

---

**Algorithm 1:** FIT algorithm on any $\text{EFO}_1$ formulas, where FITC is FIT computed on a query graph, explained in Algorithm2.

---

**Input:** $\text{EFO}_1$ query $\phi(y)$, Relation matrices $\{P_r\}_{r \in \mathcal{R}}$

1   Change the EFO1 query to DNF, $\phi_{\text{DNF}}(y) = \gamma_1(y) \vee \cdots \vee \gamma_m(y)$;

2   **for** $\gamma_i$ *in* $\phi_{\text{DNF}}(y)$ **do**

3       For the conjunctive query $\gamma_i(y)$, create its query graph $G_{\gamma_i}$;

4       Do the initialization, get $\{C_u\}_{u \in G_{\gamma_i}}$;

5       Compute each answer $A_{\gamma_i} = \text{FITC}\, (G_{\gamma_i}, \{C_u\}_{u \in G_{\gamma_i}})$;

6   **end**

7   Aggregate the sub answers and get the final answer $A[\phi(y)] = \perp_i[A_{\gamma_i}]$;

**Output:** $A[\phi(y)]$

---

The derivation is as follows:

$$A[\phi(y)](a) = T(\phi(a)) = \perp^{\star}_{x=b}[C_x(b)\top P_r(b,a)\top C_y(a)\top T(\psi(a))]$$
$$= \perp^{\star}_{x=b}[C^{\star}_y(b,a)\top T(\psi(a))]$$

where $C^{\star}_y = [(P_r \circ^{\top} C_y)^{\intercal} \circ^{\top} C_x]^{\intercal}$.

Then we found that this is the same as the situation in equation 11 which we have explained already.

To be specific, provided $\perp^{\star}$ is Godel, we have the following derivation;

$$A[\phi(y)](a) = \max_{x=b}[C^{\star}_y(b,a)\top T(\psi(a))]$$
$$= \max_{x=b}[C^{\star}_y(b,a)]\top T(\psi(a))$$

Let $C'_y(a) = \max_{x=b}[C^{\star}_y(b,a)]$ do the trick.

By the end of this step, all leaf nodes are removed.

### D.5   STEP 5: ENUMERATION ON CIRCLE

We deal with the circle in the query graph: the only technique is cutting one node $x$ and doing the enumeration: $T(\exists x.\phi(x)) = \perp^{\star}_{a \in \mathcal{E}} T(\phi(a))$.

We choose a node $u$ that cutting it can still keep the remaining graph connected, whose existence is guaranteed by graph theory (Gallier, 2011).

In practice, we can set a hyperparameter $M$ to limit the maximum number of enumerations, by the assumption 22, we can distinguish observed nodes from predicted ones. For node $u$, we sort all its candidates and enumerate $|\{a \in \mathcal{E}|C_u(a) = 1\}| + M$ the most possible ones.

In this step, we choose a node in the query graph to become a constant entity, which allows for returning back to step 3 again. In this way, the query graph becomes smaller and smaller.

### D.6   STEP 6: GETTING ANSWER VECTOR

As all steps assure the connectivity of the query graph, finally, the query graph will only contain the free variable $y$, with the only remaining formula being $\mu(y, C_y)$, then by definition, its answer vector will be $C_y$.

Additionally, we offer the pseudo-code for our FIT algorithm in Algorithm 1 and Algorithm 2. We also note that the design of the FIT algorithm is versatile and can be extended to a more complicated version (Fei et al., 2024).

## E   COMPLEXITY OF FIT ALGORITHM

We discuss the complexity of FIT algorithm here. For FIT on a general query graph, it is clearly NP-complete due to our discussion in Section 4.3. Specifically, FIT has the complexity of $O(|\mathcal{E}|^n)$,

---

**Algorithm 2:** FIT on a conjunctive query, which is represented by a query graph. We name it FITC for short.

---

1   FITC $(G, \{C_u\}_{u \in G})$
2   **if** *G contains only one node y* **then**
3     |   **return** $C_y$
4   **end**
5   **if** *G contains a node u with self loop edges.* **then**
6     |   Remove the self loop edges and changes $C_u$ simultaneously as explained in Step 2;
7   **end**
8   **if** *G contains constant entity node.* **then**
9     |   Remove the constant entity node as explained in Step 3;
10   **end**
11   **if** *G contains a leaf node u which only connects to node v* **then**
12     |   **if** *u represents free variable, v represents an existential variable.* **then**
13     |     |   Change $v$ to free variable, removing the node $u$, the new query graph is $G'$ ;
14     |     |   sub answer = FITC $(G', \{C_u\}_{u \in G'})$ ;
15     |     |   final answer is computed by sub answer as Step 4, case 2;
16     |     |   **return** *final answer*
17     |   **end**
18     |   **else if** *u represents an existential variable, v represents the free variable.* **then**
19     |     |   Update $C_v$ according to Step 4, case 3, getting $C'$ ;
20     |     |   Remove the node $u$, the query graph is $G'$ ;
21     |     |   **return** FITC $(G', \{C'_u\}_{u \in G'})$
22     |   **end**
23     |   **else**
24     |     |   Update $C_v$ according to Step 4, case 1, getting $C'$ ;
25     |     |   Remove the node $u$, the query graph is $G'$ ;
26     |     |   **return** FITC $(G', \{C'_u\}_{u \in G'})$
27     |   **end**
28   **end**
29   **else**
30     |   Find a node $u$ to enumerate, according to step 5 ;
31     |   Construct the candidate list for node $u$, namely the top N biggest index of $C_u$, where $N = M + \{a \in \mathcal{E} | C_u(a) = 1\}$ ;
32     |   Create a matrix $E \in [0,1]^{N*|\mathcal{E}|}$ to store enumerate answer ;
33     |   **for** *The ith candidate of node u, a* **do**
34     |     |   Store the original truth value of $C_u(a)$ ;
35     |     |   Change the $C_u$ to the one-hot vector $C'_u = \mathbf{1}_{\{x=a\}}$ ;
36     |     |   Change the node $u$ to represent a constant entity in query graph, creating $G'$ ;
37     |     |   Enumerate answer A = FITC $(G', C')$ ;
38     |     |   $E[i] = A * C_u(a)$
39     |   **end**
40     |   **return** $\perp_i^\star [E(i,j)]$
41   **end**

---

where n is the number of the variable in the query graph. In the worst case, the query graph has no constants and is a complete graph [2], and FIT degrades to enumeration because of circles.

However, as we have also discussed in Section 4.3, Tree-Form queries are known to be tractable and we would like to discuss FIT complexity in the TF $\cap$ EFO$_1$ queries, the result is given as the following:

**Lemma 34.** *For a query graph with $n$ variables, brutal-force implementation of FIT has a complexity of $O(n \times |\mathcal{E}|^2)$.*

---

[2]Complete graph means that every node in the graph connects to every other node in the graph.

*Proof.* We prove this lemma easily by noticing that the complexity in Step 3 and Step 4 is all we need to consider. We note that the bottleneck of the complexity comes from the computation of $C_y^\star = [(P_r \circ^\top C_y)^\intercal \circ^\top C_x]^\intercal$, which is $O(|\mathcal{E}|^2)$, while the other computations have no larger complexity. Because it takes $O(|\mathcal{E}|^2)$ to remove one node in the query graph, the total complexity is $O(n \times |\mathcal{E}|^2)$. $\square$

**Proposition 35.** *For a query graph with $n$ variables, an efficient implementation of FIT has a complexity of $O(n \times t^2)$, where t is a coefficient defined by the sparsity of the matrix.*

$$t = \max_{r \in \mathcal{R}} |\{a \in \mathcal{E} | \exists b. P_r(a, b) > 0 \text{ or } P_r(b, a) > 0\}| \tag{12}$$

*Proof.* The proof is straightforward by the same proof technique used in Appendix A.2. By the definition of t, every matrix of relation $r$ can be simplified as an $t * t$ dense matrix, and every $C_u$ in the computation process is guaranteed to have no more than $t$ elements that are larger than 0. Then, the optimal complexity of FIT is $O(n \times t^2)$. $\square$

## F    TRAINING AND TESTING DETAILS

We utilize pretrained neural link predictors provided by Minervini et al. (2022), who followed previous knowledge graph embeddings method (Trouillon et al., 2016) which gives a score $s(h, a, b)$ for every possible triple$(a, r, b)$, where $a, b \in \mathcal{E}, r \in \mathcal{R}$. To convert real number scores to truth value that falls into $[0, 1]$, we use the softmax function:

$$P_{r,a}^\star(b) = \frac{exp(s(a, r, b))}{\Sigma_{c \in \mathcal{E}} exp(s(a, r, c))}$$

However, we notice that this implies that there's only one tail node for $(a, r)$, thus we compute the scaling factor by utilizing the observed edges in the observed graph $\mathcal{G}_o$. We define the set $E_{a,r} = \{b \mid (a, r, b) \in \mathcal{G}_o\}$.

$$Q_{a,b} = \begin{cases} \frac{|E_{a,r}|}{\Sigma_{c \in E_{a,r}} P_{r,a}^\star(c)}, & \text{if } |E_{a,r}| > 0 \\ 1, & \text{if } |E_{a,r}| = 0 \end{cases}$$

For training, we just clamp the value for each triple:

$$P_r(a, b) = min(1, P_{r,a}^\star(b) * Q_{a,b})$$

However, in testing, we compute the final truth value by combining the computed probability with the required property in Assumption 22:

$$P_r(a, b) = \begin{cases} 1, & \text{if } b \in E_{a,r} \\ 0, & \text{if } P_{r,a}^\star(b) * Q_{a,b} < \epsilon \\ min(P_{r,a}^\star(b) * Q_{a,b}, 1 - \delta), & \text{otherwise} \end{cases} \tag{13}$$

We note that $\epsilon, \delta$ are hyper-parameters, $\epsilon$ acts as a threshold which ensures for every $r$, $P_r$ is a sparse matrix and $\delta > 0$ is required to meet our Assumption 22.

Once the $P_r$ is a sparse matrix, our complexity discussion in Appendix E applies. However, there are some query types that do not require the matrix to be sparse, namely the 1p, 2i, 3i, 2in, and 3in, since there is no existential variable in them. Therefore, the computation becomes simple as we only need to compute one row in the whole matrix.

Therefore, when dealing with these five query types, we just use the dense matrix:

$$\overline{P}_r(a, b) = \begin{cases} 1, & \text{if } b \in E_{a,r} \\ min(P_{r,a}^\star(b) * Q_{a,b}, 1 - \delta), & \text{otherwise} \end{cases} \tag{14}$$

For the training, we only use 1p, 2i, 3i, 2in, and 3in query types for efficiency, the reason is explained above. The learning rate is set to 0.0001, the batch size is set to 64, the maximum training step is set to 5,000 steps and we choose the best checkpoints by the scores in the validation set. The choice of the hyperparameter is in Appendix H.

## G CONNECTIONS TO PREVIOUS METHODS

In this section, we discussion the connections between FIT and previous methods.

### G.1 CONNECTION TO TRADITIONAL METHOD

It has been traditionally found that acyclicity is the key to tractability: researchers from two different backgrounds, namely the databases theory (Yannakakis, 1981) and constraint satisfaction problem (Dechter & Pearl, 1987) has all found out that conjunctive queries can be evaluated in polynomial time if it is acyclic. However, their results are both in classical settings, where the truth value can either be 0 or 1, while our setting is extremely general, not only do we have a probabilistic value for each triple, but our Definition 18 is the most general one which allows for two arbitrary t-norms be included in the computation. Moreover, we offer efficient computation that is purely vectorized, which is provided in detail in Appendix D. Most significantly, FIT is a neural-symbolic method, the neural link predictor can be fine-tuned by complex query to further address the OWA, while the traditional methods can never **predict** any answer that is not observed in $\mathcal{G}_o$, which is why machine learning techniques should step in.

### G.2 CONNECTIONS TO QTO

We note that QTO (Bai et al., 2023) is only a simplified version of FIT, in other words, FIT is a natural extension of QTO in the sense that FIT is capable of solving $\text{EFO}_1$ queries that QTO fails to represent syntactically because QTO relies on the operator-tree method. Moreover, please see the proposition below for the inference of queries within $\text{TF} \cap \text{EFO}_1$.

**Proposition 36.** *FIT can coincide with QTO when answering queries within the* $\text{TF} \cap \text{EFO}_1$ *family.*

*Proof.* Since QTO also converts all formulas in DNF, we are allowed to consider only conjunctive queries. Consider a conjunctive query

$$\gamma = \exists x_1, \cdots, x_n.\alpha_1 \wedge \cdots \wedge \alpha_m$$

where $\alpha_i$ is and atomic formula or its negation. Then, we note that for the objectives of QTO, it aims to find free variable $y$ such that it maximizes:

$$\phi(y) = \max_{x_1, \cdots, x_n} T(\alpha_1)\top \cdots \top T(\alpha_m) = \bot^{\star}_{x_1, \cdots, x_n}[T(\alpha_1)\top \cdots \top T(\alpha_m)] \tag{15}$$

where the $\bot^{\star}$ is Godel t-conorm, and the truth of value is negation is also defined by $T(\neg\phi) = 1 - T(\phi)$ in QTO (Bai et al., 2023). By the analysis, we show that these optimization objectives naturally coincide with the definition of truth value we proposed systematically in Definition 18. Thus, QTO and FIT yield the same result in $\text{TF} \cap \text{EFO}_1$ queries as long as these requirements are met: 1. The same learning-base matrix is given with no further fine-tuning, 2. FIT chooses Godel to be its existential $t$-conorm and product to be its conjunctive $t$-norm, and 3: In queries with no existential variable, FIT also uses a sparse matrix rather than the dense ones. $\square$

We have also done experiments to verify this proposition, we show the result in Table 4, in which we show they coincide with each other. In this way, we show that QTO is at best a special case of FIT, and FIT serves as a superior alternative to QTO in all $\text{EFO}_1$ queries.

## H HYPERPARAMETER IMPACT

In this section, we discuss several hyperparameters used in the FIT algorithm.

Table 4: The MRR results(%) of FIT versus QTO on the TF $\cap$ EFO$_1$ queries (BetaE dataset) on FB15k-237.

| Method | 1p | 2p | 3p | 2i | 3i | ip | pi | 2u | up | 2in | 3in | inp | pin | AVG. |
|---|---|---|---|---|---|---|---|---|---|---|---|---|---|---|
| FIT - QTO | 0 | 0 | 0 | 0 | 0 | 0 | 0 | 0 | 0 | 0 | 0 | 0 | 0 | 0 |

Table 5: Matrix related hyper parameter impact on the MRR results(%) on the BetaE dataset on FB15k-237.

| $\epsilon, \delta$ | 1p | 2p | 3p | 2i | 3i | ip | pi | 2in | 3in | inp | pin | 2u | up | AVG. |
|---|---|---|---|---|---|---|---|---|---|---|---|---|---|---|
| 0.002, 0.001 | **46.70** | **14.65** | **12.87** | 37.52 | **51.55** | **22.19** | **30.46** | **13.99** | 19.99 | **10.18** | 9.54 | **18.04** | 13.09 | **23.14** |
| 0.005, 0.001 | **46.70** | 14.61 | 12.80 | 37.52 | **51.55** | 21.88 | 30.10 | **13.99** | 19.99 | **10.18** | **9.55** | **18.04** | 13.08 | 23.08 |
| 0.005, 0.01 | **46.70** | 14.62 | 12.80 | **37.54** | 51.54 | 21.88 | 30.10 | **13.99** | **20.01** | 10.17 | 9.53 | **18.04** | **13.09** | 23.08 |
| 0.01, 0.001 | **46.70** | 14.49 | 12.74 | 37.52 | **51.55** | 21.57 | 29.37 | **13.99** | 19.99 | 10.17 | 9.51 | **18.04** | 13.07 | 22.98 |

For the threshold $\epsilon$, we use 0.005 for both FB15k-237 and FB15k, 0.0002 for NELL. For the $\delta$, it is 0.001 in all datasets. For $M$, it is 10 in all datasets. As for the tnorm, We choose Godel (maximum) to represent the existential quantifier as discussed in equation 11, and we choose the product to represent logic conjunction because of its clear probability interpretation, which also follows the previous research (Arakelyan et al., 2020). Then we study the impact of each hyper-parameter one by one.

Here we present the impact of the hyperparameter, the result is shown in Table 5.

As shown, the impact of $\delta$ is vert marginal, because very few predicted answers have comparable scores with the observed answers. As for the $\epsilon$, the smaller, the better, which is kind of obvious because the larger threshold monotonically leads to a sparser matrix that loses more information about the sorting of the predicted tail variables. Moreover, we note when inferring 1p,2i,3i,2in, and 3in queries, we do not use the sparse matrix, therefore $\epsilon$ have no impact on these 5 query types.

Then we also discuss the impact of setting the maximum number of enumerations, the result is shown in Table 6, where we can see that bigger $M$ leads to better performance, which is intuitively true because taking more possible candidates for inner variables into consideration is beneficial for getting the final answer.

Finally, we show that it's also possible to choose other $t$-norms for the conjunction, using Godel as the conjunction $t$-norm also has its own advantage in some queries though it is slightly worse than the product on average, shown in Table 7. However, as our computation in equation 9 shows, using Godel as the existential $t$-norm is the only practical way for allowing FIT to compute efficiently.

## I  SELF LOOP

Though this circumstance is allowed in logic and explained in our Section 5, we find that it is rare in real-world KG like NELL. In fact, there are only four relations in NELL that have self-loop. In FB15k and FB15k-237, most of the self-loop triples are redundant like "New York is located at New York". Thus, we do not sample formulas with self-loop, which is also inherited by other work (Yin et al., 2023; 2024).

## J  DATASET STATISTICS

The number of queries in the newly developed real EFO$_1$ dataset is shown in Table 8. We note that the "pni" data retains the previous queries sampled by Ren & Leskovec (2020), thus, we retain the number of queries to be different in different datasets. For those brand new queries proposed by ourselves, we sampled 5000 for each query type in each dataset.

For the formal first order formulas of our newly developed real EFO$_1$ dataset, we also offer them in the following Table 9.

Table 6: Max enumeration impact on the MRR results(%) on the circle query on FB15k-237.

| M | 3c | 3cm | AVG. |
|---|---|---|---|
| 5 | 39.2 | 37.1 | 38.2 |
| 10 | 39.4 | 37.3 | 38.4 |
| 20 | 39.5 | 37.5 | 38.5 |

Table 7: The impact of changing conjunction $t$-norm on the MRR results(%) on FB15k-237.

| Method | Query Type | pni | 2il | 3il | 2m | 2nm | 3mp | 3pm | im | 3c | 3cm | AVG. | | | |
|---|---|---|---|---|---|---|---|---|---|---|---|---|---|---|---|
| Product | EFO1 | **14.9** | **34.2** | **51.4** | **9.9** | **12.7** | **19.6** | **11.9** | **7.7** | **39.4** | **37.3** | **23.9** | | | |
| Godel | | 14.2 | 33.6 | 49.3 | 8.4 | 11.8 | 14.9 | 10.7 | 6.0 | 35.5 | 33.3 | 21.8 | | | |

| Method | Query Type | 1p | 2p | 3p | 2i | 3i | ip | pi | 2in | 3in | inp | pin | 2u | up | AVG. |
|---|---|---|---|---|---|---|---|---|---|---|---|---|---|---|---|
| Product | Tree-Form | **46.7** | **14.6** | **12.8** | **37.5** | **51.6** | **21.9** | **30.1** | 14.0 | 20.0 | **10.2** | **9.5** | **18.0** | **13.1** | **23.1** |
| Godel | | **46.7** | 14.1 | 12.0 | 35.1 | 44.7 | 19.7 | 28.3 | **14.1** | **20.3** | 10.0 | **9.5** | 17.9 | 12.5 | 21.9 |

## K    BASELINE IMPLEMENTATION

In this section, we illustrate the implementation of the baseline method, as we have explained in Section 3, previous methods can not directly represent our newly developed dataset. One example in Figure 6 illustrates how the Tree-Form queries can **approximate** our newly developed queries to their best.

For those methods that depend on the operator tree, which include BetaE (Ren & Leskovec, 2020), LogicE (Luus et al., 2021), ConE (Zhang et al., 2021), and QTO (Bai et al., 2023). similarly with the example in Figure 6, we manually create a new operator tree to approximate each of our new formulas, which is explained in Table 10 and Table 11. Because of the limitation of the operator tree form, these formulas are only approximations, we also offer the real logical formula it represents, making the difference with the real $EFO_1$ formulas in Table 9 more apparent. For the method that utilizes query graph (Minervini et al., 2022; Wang et al., 2023b), we use our query graph with their provided checkpoint directly.

Table 8: Statistics of our new proposed real $EFO_1$ dataset.

| Knowledge Graph | pni | 2il | 3il | 2m | 2nm | 3mp | 3pm | im | 3c | 3cm |
|---|---|---|---|---|---|---|---|---|---|---|
| FB15k-237 | $5 \times 10^3$ | $5 \times 10^3$ | $5 \times 10^3$ | $5 \times 10^3$ | $5 \times 10^3$ | $5 \times 10^3$ | $5 \times 10^3$ | $5 \times 10^3$ | $5 \times 10^3$ | $5 \times 10^3$ |
| FB15k | $8 \times 10^3$ | $5 \times 10^3$ | $5 \times 10^3$ | $5 \times 10^3$ | $5 \times 10^3$ | $5 \times 10^3$ | $5 \times 10^3$ | $5 \times 10^3$ | $5 \times 10^3$ | $5 \times 10^3$ |
| NELL | $4 \times 10^3$ | $5 \times 10^3$ | $5 \times 10^3$ | $5 \times 10^3$ | $5 \times 10^3$ | $5 \times 10^3$ | $5 \times 10^3$ | $5 \times 10^3$ | $5 \times 10^3$ | $5 \times 10^3$ |

Table 9: Formal definition of the new proposed dataset, where $a_i \in \mathcal{E}$ are entities from knowledge graph, $x_i$ are bounded existential variables, and $y$ is the single free variable.

| Name | $EFO_1$ Formula |
|---|---|
| pni | $\exists x.r_1(a_1, x) \wedge \neg r_2(x, y) \wedge r_3(a_2, y)$ |
| 2il | $\exists x.r_1(a, y) \wedge r_2(x, y)$ |
| 3il | $\exists x.r_1(a_1, y) \wedge r_2(a_2, y) \wedge r_3(x, y)$ |
| 2m | $\exists x.r_1(a, x)) \wedge r_2(x, y) \wedge r_3(x, y)$ |
| 2nm | $\exists x.r_1(a, x) \wedge r_2(x, y) \wedge \neg r_3(x, y)$ |
| 3mp | $\exists x_1, x_2.r_1(a, x_1) \wedge r_2(x_1, x_2) \wedge r_3(x_2, y) \wedge r_4(x_1, x_2)$ |
| 3pm | $\exists x_1, x_2.r_1(a, x_1) \wedge r_2(x_1, x_2) \wedge r_3(x_2, y) \wedge r_4(x_2, y)$ |
| im | $\exists x.r_1(a_1, x) \wedge r_2(a_2, x) \wedge r_3(x, y) \wedge r_4(x, y)$ |
| 3c | $\exists x_1, x_2.r_1(a_1, x_1) \wedge r_2(x_1, y) \wedge r_3(a_2, x_2) \wedge r_4(x_2, y) \wedge r_5(x_1, x_2)$ |
| 3cm | $\exists x_1, x_2.r_1(a_1, x_1) \wedge r_2(x_1, y) \wedge r_3(a_2, x_2) \wedge r_4(x_2, y) \wedge r_5(x_1, x_2) \wedge r_6(x_1, y)$ |

Table 10: The lisp-like formula for the approximation of our newly developed dataset, it is invented by Wang et al. (2021) to express any kind of Tree-Form query.

| Name | Lisp-like formula |
|---|---|
| pni | (i,(n,(p,(p,(e)))),(p,(e))) |
| 2il | (p,(e)) |
| 3il | (i,(p,(e)),(p,(e))) |
| 2m | (i,(p,(p,(e))),(p,(p,(e)))) |
| 2nm | (i,(n,(p,(p,(e)))),(p,(p,(e)))) |
| 3mp | (p,(i,(p,(p,(e))),(p,(p,(e))))) |
| 3pm | (i,(p,(p,(p,(e)))),(p,(p,(p,(e))))) |
| im | (i,(p,(i,(p,(e)),(p,(e)))),(p,(i,(p,(e)),(p,(e))))) |
| 3c | (i,(p,(i,(p,(e)),(p,(p,(e))))),(p,(p,(e)))) |
| 3cm | (i,(i,(p,(p,(e))),(p,(p,(e)))),(p,(i,(p,(e)),(p,(p,(e)))))) |

Table 11: The formal first order logic formula for the approximation of our newly developed dataset.

| Name | Tree Form formula |
|---|---|
| pni | $\forall x.r_2(b, y) \wedge (\neg r_1(x, y) \vee \neg r_3(a, x))$ |
| 2il | $r(a, y)$ |
| 3il | $r_1(a, y) \wedge r_2(b, y)$ |
| 2m | $\exists x_1, x_2.r_1(a, x_1) \wedge r_2(x_1, y) \wedge r_1(a, x_2) \wedge r_3(x_2, y)$ |
| 2nm | $\exists x_1, \forall x_2.r_1(a, x_1) \wedge r_2(x_1, y) \wedge (\neg r_1(a, x_2) \vee \neg r_3(x_2, y))$ |
| 3mp | $\exists x_1, x_2, x_3.r_1(a, x_1) \wedge r_2(x_1, x_2) \wedge r_1(a, x_3) \wedge r_4(x_3, x_2)) \wedge r_3(x_2, y)$ |
| 3pm | $\exists x_1, x_2, x_3, x_4.r_1(a, x_1) \wedge r_2(x_1, x_2) \wedge r_3(x_2, y) \wedge r_1(a, x_3) \wedge r_2(x_3, x_4) \wedge r_4(x_4, y)$ |
| im | $\exists x_1, x_2.r_1(a_1, x_1) \wedge r_2(a_2, x_1) \wedge r_3(x_1, y) \wedge r_1(a_1, x_2) \wedge r_2(a_2, x_2) \wedge r_4(x_2, y)$ |
| 3c | $\exists x_1, x_2, x_3.[r_1(a_1, x_1) \wedge r_2(x_1, y)] \wedge [r_3(a_2, x_2) \wedge r_1(a_1, x_3) \wedge r_5(x_3, x_2) \wedge r4(x_2, y)]$ |
| 3cm | $\exists x_1, x_2, x_3, x_4.[r_1(a_1, x_1) \wedge r_2(x_1, y) \wedge r_1(a_1, x_4) \wedge r_6(x_4, y)] \wedge r_3(a_2, x_2) \wedge r_1(a_1, x_3) \wedge r_5(x_3, x_2) \wedge r4(x_2, y)$ |

