# OpenReview forum: "Rethinking Complex Queries on Knowledge Graphs with Neural Link Predictors"
_ICLR.cc/2024/Conference — ICLR 2024 poster_

### Official Review · Reviewer_QmUh · 2023-10-30

**Soundness:** 4 excellent
**Presentation:** 4 excellent
**Contribution:** 4 excellent
**Rating:** 8
**Confidence:** 4

**Summary:**

The paper addresses the challenging task of reasoning on knowledge graphs using neural link predictors. The prevailing method in this field is query embedding, which treats logic operations as set operations and has shown empirical success. However, the paper argues that many claims made in previous research lack a formal and systematic inspection. To address this, the authors characterize the scope of previously investigated queries, identify the gap between the formulation and the goal, and provide complexity analysis for the queries. They also introduce a new dataset with ten new types of queries and propose a new neural-symbolic method called Fuzzy Inference with Truth value (FIT) that combines neural link predictors with fuzzy logic theory. The empirical results demonstrate that FIT outperforms previous methods significantly in the new dataset and also surpasses previous methods in the existing dataset.

**Strengths:**

1. The paper shows many different opinions against previous series of works. The discussion firmly supports that TF query family is not even a subset of EFO1. In addition, the paper rethinks the traditional claim that “reasoning involves an exponential growth in computational time”. The proposition and analysis finally found that the Tree-Form EFO1 reasoning complexity is linear to the number of variables in the query.
2. The paper provides a comprehensive analysis of the prevailing method of query embedding and its limitations. It identifies the gap between the formulation and the goal, which adds clarity to the field and helps in understanding the limitations of existing approaches.
3. The introduction of a new dataset with ten new types of queries is a significant contribution. This dataset allows for a thorough investigation of complex queries and provides a benchmark for evaluating future methods.
4. The proposed method, FIT, which combines neural link predictors with fuzzy logic theory, is a novel approach. It addresses the limitations of previous methods and demonstrates improved performance in both the new and existing datasets.
5. The paper presents empirical results that support the superiority of FIT over previous methods. The significant improvement in performance in both datasets strengthens the credibility of the proposed method.

**Weaknesses:**

Lack of introduction to related works: The paper lacks a section on related works, making it difficult for readers who are not familiar with the definitions and concepts in previous research. This can hinder the understanding of the paper and limit its accessibility to a broader audience.

**Questions:**

The paper could benefit from providing more examples and illustrations to aid in understanding the concepts and methodologies discussed.

---

> ### Author Response · Authors · 2023-11-17
>
> Dear reviewer, we thank you for your insightful comments and here is our response.
>
> > Lack of introduction to related works: The paper lacks a section on related works, making it difficult for readers who are not familiar with the definitions and concepts in previous research. This can hinder the understanding of the paper and limit its accessibility to a broader audience.
>
> We appreciate your valuable advice, we have noted that our paper somewhat assumes the readers to have some background in the development of complex query answering. We will add a new section of related work in the Appendix in the camera-ready version according to your suggestions.
>
>
> > The paper could benefit from providing more examples and illustrations to aid in understanding the concepts and methodologies discussed.
>
> Thank you for your suggestions. We will try our best to include more intuitive illustrations for better understanding in the camera-ready version, especially at the methodology part.

---

### Official Review · Reviewer_Z6xA · 2023-11-01

**Soundness:** 3 good
**Presentation:** 3 good
**Contribution:** 3 good
**Rating:** 8
**Confidence:** 3

**Summary:**

This paper shares author's rethinking over complex queries on knowledge graph with neural link predictor especially towards answering EFO_1 queries. Authors discussed the difference between previous widely researched tree-form queries and EFO_1 queries and their relationships, and pointed out that the tree-form queries are with unrigorous formulations. Thus they propose to pull the complex query answering research back to answering EFO_1 queries with rigorous formulations, and propose a method with neural link predictor based on fuzzy logic theory, called FIT. FIT outperforms baselines significantly in the new EFO_1 query dataset and also surpasses baselines in the existing dataset at the same time.

**Strengths:**

1. Authors rethink complex queries on KGs with theoretical analysis.
2. A new EFO_1 dataset are created to evaluate the EFO_1 query answering task.
3. And the proposed method FIT shows good performance over EFO_1 datasets and existing tree-form datasets.

**Weaknesses:**

The basic idea of the new proposed method is based on fuzzy logics and neural link predictor. This general idea is also introduced in FuzzQE[1]. But the key difference between FIT and FuzzQE is not clearly discussed and FuzzQE is not compared in Table 2.

[1] Xuelu Chen, Ziniu Hu, Yizhou Sun. Fuzzy Logic Based Logical Query Answering on Knowledge Graphs.  AAAI2022.

**Questions:**

1. Should the $r_1$ and $r_2$ be exchanged in Equation (3)? i.e. Should Equation (3) be $\forall x. (r_3(b, y) ∧ ¬r_2(x, y)) ∨ (r_3(b, y) ∧ ¬r_1(a, x))$ ?
2. What is main difference between FuzzQE and FIT?

---

> ### Author Response · Authors · 2023-11-17
>
> Dear reviewer, we thank you for your insightful comments and here is our response.
>
>
> >  The basic idea of the new proposed method is based on fuzzy logics and neural link predictor. This general idea is also introduced in FuzzQE[1]. But the key difference between FIT and FuzzQE is not clearly discussed and FuzzQE is not compared in Table 2.
>
>
> We thank you for mentioning FuzzQE. FuzzQE is surely an interesting paper that leverages fuzzy logic theory to build its model and we are aware of its contribution and have already cited this paper in our paper. However, as we have already stated in the whole Section 3 and Section 4, the previous method, including FuzzQE, only can represent Tree-Form query, and is therefore fundamentally different from our proposed FIT which can represent any EFO1 query. Therefore, FuzzQE is actually similar to other query embedding methods, like ConE[2] and BetaE[1], and we note the performance of FuzzQE is also similar to ConE[2] as reported in their paper, therefore, we do not compare FIT with FuzzQE in detail. Regarding your question about Table 2, all baseline methods we discuss here share the same pretrained KGE checkpoints, while FuzzQE does not. Therefore, we do not include FuzzQE in the Table 2 as it is not suitable for our discussion.
>
> We will consider including it as a baseline in Table 1 instead in the camera-ready version.
>
>
> [1]Ren, Hongyu, and Jure Leskovec. "Beta embeddings for multi-hop logical reasoning in knowledge graphs." Advances in Neural Information Processing Systems 33 (2020): 19716-19726.
>
> [2]Zhang, Zhanqiu, et al. "Cone: Cone embeddings for multi-hop reasoning over knowledge graphs." Advances in Neural Information Processing Systems 34 (2021): 19172-19183.
>
>
>
> > Should the formula be exchanged in Equation (3)?
>
>
> Sorry for this typo, we have changed it based on your advice.
>
>
> > What is main difference between FuzzQE and FIT?
>
> Please check our rebuttal of the first question.

---

> > ### Comment · Reviewer_Z6xA · 2023-11-22
> > **Acknowledgement**
> >
> > Thanks for clarification. I do not have further question.

---

### Official Review · Reviewer_TtwJ · 2023-11-07

**Soundness:** 2 fair
**Presentation:** 3 good
**Contribution:** 4 excellent
**Rating:** 6
**Confidence:** 4

**Summary:**

The paper has 2 main contributions:

1. Describing the gaps in EFO1 queries studied by past work
- Past work [1] has studied a subset of existential first-order logic queries on knowledge graphs (KGs). In particular, past work has studied tree-form queries with negated atomic relations
- Authors point out that the studied query types do not entirely cover the EFO1 query family and describe the missing families of query structures
- They create a dataset Real-EFO1 that consists of EFO1 query structures not studied by past work and benchmark competitive graph query execution approaches

2. Proposing a fuzzy-logic based approach for executing EFO1 queries on knowledge graphs
- Authors propose a fuzzy-logic based query execution algorithm Fuzzy Inference with Truth values (FIT)
- The algorithm extends QTO proposed by [2] in the following ways:
    - It extends QTO to handle cycles in the query graph
    - It sparsifies the neural link prediction matrices to improve the run-time complexity of the algorithm for certain query types
    - It demonstrates that the link predictor can be trained end-to-end directly with the query execution training data

[1] Hongyu Ren and Jure Leskovec. Beta embeddings for multi-hop logical reasoning in knowledge graphs. NeurIPS 2020
[2] Yushi Bai, Xin Lv, Juanzi Li, and Lei Hou. Answering Complex Logical Queries on Knowledge Graphs via Query Computation Tree Optimization. ICML 2023

**Strengths:**

1. The paper clearly describes differently query families and establishes the query structures that are missing from past work
2. The proposed Real-EFO1 dataset extends the family of query structures studied in the query execution literature with intuitive examples and connections to past work
3. The proposed FIT approach shows consistent improvement over the baseline approaches and is properly ablated
    - Experiments show that FIT reduces to QTO under the appropriate conditions and that the additional differences lead to performance improvements across all settings
    - Examples show how past approaches (designed for tree-form queries) cannot easily handle the new query structures while fuzzy inference can handle them

**Weaknesses:**

1. I believe that the paper misunderstands the definition of tree-form queries used by [1]
    - [1] explicitly limit their definition of tree-form queries to only consider the negation of individual atomic formulae (i.e. they consider queries can by build from a query tree using $r(x, y)$ and $\neg r(x, y)$. Therefore, the problem of introducing universal quantifiers is avoided by [1]
    - However, definition 8 in this paper uses a different (broader) definition of tree-form queries. Under this new definitions, tree-form queries could require universal quantification and fall outside the EFO1 category
    - I believe this mismatch shakes some of the grounded for Section 3 (Section 4 is still valid in my opinion, since it tries to describe EFO1 - TF)
2. The connections between FIT and QTO are not sufficiently stressed in the main paper (and this analysis is pushed to the Appendix)
    - I believe that making this connection is important for a fair comparison to QTO

[1] Hongyu Ren and Jure Leskovec. Beta embeddings for multi-hop logical reasoning in knowledge graphs. NeurIPS 2020

**Questions:**

Questions
---
1. I would like to reiterate Weakness 1 described above. I believe that the definition of tree-form queries used in this work differs from the definition used by past work and that the mismatch makes Section 3 obsolete. Please comment.

Typos
---
1. (Important) Definition 6: Missing definition of EFO1 query family

Suggestions
---
1. Eq 4: It would be good to have an intuitive definition of k here
2. Overall, the paper pushes a lot of context to the Appendix. It would be helpful if the main paper summarized the corresponding Appendix section rather than just point to it

---

> ### Author Response · Authors · 2023-11-17
>
> We thank you for your time and review. However, there seem to be some core misunderstandings about our major claim that we would like to clarify.
>
> Regarding your first problem:
>
>
> > I believe that the paper misunderstands the definition of tree-form queries used by [1]. [1] explicitly limit their definition of tree-form queries to only consider the negation of individual atomic formulae.
>
> We thank you for stating your concerns directly. It is true that [1] has claimed to consider negation of atomic formula, and avoid universal quantifier in their preliminary. However, below the Definition 8, we have stated that the query types they actually investigated (Tree-Form)  deviate from their original goal (EFO1). Specifically, we state that the definition of Tree-form query is from direct derivation of the query embedding methods utilized in [1], we also include this part of the proof in Appendix B. We welcome you to check that. Moreover, we also welcome you to check the “pni’’ data explained in our Example 10, which is available in the dataset provided by [1]. We note that we have again considered the new ``pni’’ data in our new dataset, because it is treated as the universal version in [1] but as the existential version in our paper. To summarize, we are actually inheriting the claim of [1] to infer any EFO1 query, and at the same time discussing what is wrong or missing in the previous pratice.
>
> [1] Hongyu Ren and Jure Leskovec. Beta embeddings for multi-hop logical reasoning in knowledge graphs. NeurIPS 2020
>
>
> > The connections between FIT and QTO are not sufficiently stressed in the main paper (and this analysis is pushed to the Appendix). I believe that making this connection is important for a fair comparison to QTO
>
>
> Thank you for your advice. We have included the conclusion that FIT is an extension of QTO in the main paper now according to your advice and then point to the appendix for the detailed proof. We are sorry that the space of the paper is restricted for more detail and we will try our best to put every conclusion in the main paper while leaving details and proofs in the appendix.
>
> > (Important) Definition 6: Missing definition of EFO1 query family
>
> Sorry for this typo, Definition 6 should only be the answer set of the EFO1 query, as the EFO1 query has already been defined before Definition 4. We have corrected this typo according to your advice.
>
> > Eq 4: It would be good to have an intuitive definition of k here
>
> We thank you for this suggestion, we have restated that k is a coefficient to characterize the sparsity of the knowledge graph.
>
> > Overall, the paper pushes a lot of context to the Appendix. It would be helpful if the main paper summarized the corresponding Appendix section rather than just point to it
>
>
> We will make more summarization of the main point in our main paper, for example, the connection to QTO as stated before. We value your suggestion and we thank you in advance if there is another place that you think that adding more summarization will be beneficial to our presentation.

---

> > ### Author Response · Authors · 2023-11-21
> > **Looking forward to reply**
> >
> > Dear reviewer, we wonder whether our rebuttal addresses your concern and whether there are any further suggestions or questions. We are looking forward to your reply.

---

### Official Review · Reviewer_sXg1 · 2023-11-08

**Soundness:** 3 good
**Presentation:** 2 fair
**Contribution:** 2 fair
**Rating:** 6
**Confidence:** 3

**Summary:**

The paper addresses neuro-symbolic execution of knowledge graph queries. The authors use a graph representation of a symbolic (first order logic) query, and for potentially incomplete knowledge graphs, the authors propose FIT, an algorithm to compute the answer to a query in a message-passing type of algorithm over the query graph, with probabilities of all possible relations. They argue that tree-form queries cannot represent queries with existential leaves, so they cannot cover the full space of first order logic. They show experiments where FIT outperforms existing query graph execution approaches on 10 sampled query types from 3 KGs.

**Strengths:**

1. For incomplete knowledge graphs, proposed algorithm FIT is essentially a custom message-passing algorithm over nodes and edges computing the probability of every possible relation and updating the neighboring node probabilities accordingly to reach the answer set. It may have the advantage of staying faithful to the allowed relations or combinations from the query graph and KG, since it builds the solution by exploring only those combinations, and using back-propagation to update the values from the actual answer.

2. The experiments confirm improvements over existing first order logic neuro-symbolic methods for computing the answer set.

**Weaknesses:**

1. Baselines - Graph neural networks (based on message passing and backprop over the query graph) is likely a baseline to consider, since this follows a more controlled message passing approach to the solution. This was not considered in the paper. The overall performance in mean reciprocal rank is still low (~30) in two of the datasets. What would explain the difficulty or effectiveness in those cases?

2. Presentation - The computation of the Cu(node) from the probability of the relations, is not provided in the paper (It should be in the main paper, not the appendix, to allow for key aspects to be presented as a whole). The introduction could be clearer about how or why this fuzzy approach (or neural symbolic approach) is required for incomplete KGs. Are there other benefits of it?

3. The proof for tree-form not capturing all first order logic queries I am not sure about. The authors suggest that existential leaves cannot be represented, but that does not prove that any FOL query could not be converted into a potentially different tree form structure.

**Questions:**

See weaknesses section.

---

> ### Author Response · Authors · 2023-11-17
>
> We thank you for your time and review, However, there seem to be some misunderstandings that we would like to clarify.
>
> Regarding your first problem:
>
> > Baselines - Graph neural networks (based on message passing and backprop over the query graph) is likely a baseline to consider, since this follows a more controlled message passing approach to the solution. This was not considered in the paper.
>
> We thank you for your advice on the baselines. In fact, LMPNN[1], one of our baselines, has the title of ``Logical Message Passing Networks with One-hop Inference on Atomic Formulas’’, and is a method that is based on message passing and updates the embedding of each node in the query graph in each interaction. We think this may help with your question.
>
> Regarding your problem:
>
> > The overall performance in mean reciprocal rank is still low (~30) in two of the datasets. What would explain the difficulty or effectiveness in those cases?
>
> This reflects the inherent difficulties in these two knowledge graphs, the performance of simple link prediction tasks in these two graphs is also more difficult as shown in many previous research[2,3]. Moreover, the relative performance of FIT against previous methods rather than the absolute performance should be taken into account as we already surpass all baselines in all ten query types in Table 1. Finally, we want to stress that having around 30 percent MRR is not a very low score, as it indicates that every answer is ranked third among tens of thousands of entities.
>
>
> > Presentation - The computation of the Cu(node) from the probability of the relations, is not provided in the paper (It should be in the main paper, not the appendix, to allow for key aspects to be presented as a whole). The introduction could be clearer about how or why this fuzzy approach (or neural symbolic approach) is required for incomplete KGs. Are there other benefits of it?
>
> We thank you for your advice that the computation details should be included in the main paper. In general, we aim to summarize the important points in the main paper and leave the details to the appendix. Regarding the methodology, we have given an example shown in Figure 4 and explained in Section 5.3 to help the audience understand, specifically, this example has included all three features that should be taken into account according to our Theorem 16: existential leaf, multigraph, and cyclic graph. Moreover, Section 5.4 has clearly given some theoretical guarantees (Theorem 21 and Theorem 24), they are all benefits for our fuzzy method. The detail of the computation is in the Appendix and takes 4 pages to finish, therefore, it is challenging for us to provide that in the main paper, and we will try hard to include more discussion for that in the camera-ready version.
>
> >The proof for tree-form not capturing all first order logic queries I am not sure about. The authors suggest that existential leaves cannot be represented, but that does not prove that any FOL query could not be converted into a potentially different tree form structure.
>
> The whole family of tree-form queries has been defined by Definition 8 which characterizes the expressiveness of the operator tree method, therefore, mathematically speaking, it is impossible to construct other different tree-form queries. Moreover, Theorem 16 and Figure 5 together illustrate that apart from the existential leaves that you mention, multigraph, and cyclic graph can not be represented by tree-form queries, which we believe to be more intuitive for readers to understand.
>
> [1]Zihao Wang, Yangqiu Song, Ginny Wong, and Simon See. Logical Message Passing Networks with One-hop Inference on Atomic Formulas. In The Eleventh International Conference on Learning Representations, 2023.
>
> [2]Théo Trouillon, Johannes Welbl, Sebastian Riedel, Éric Gaussier, and Guillaume Bouchard. Complex embeddings for simple link prediction. In International conference on machine learning, pp. PMLR, 2016.
>
> [3]Li, Rui, et al. "House: Knowledge graph embedding with householder parameterization." International Conference on Machine Learning. PMLR, 2022.

---

> > ### Comment · Reviewer_sXg1 · 2023-11-18
> > **acknowledgement**
> >
> > I have read the reviews - I don't have further questions. MRR at 30 still means the method very often gets the answer ranked near three, but probably barely at rank 1. For a query to be executed and be useful realiably it should be much higher.
> >
> > For the proof, the justification may not be fully sound. A reformulation of a query graph, that avoids existential leaves but can be represented within the tree form definition may be possible. The proof does not show the contradiction that it is never possible with any possibly tricky reformulation.

---

> > > ### Author Response · Authors · 2023-11-18
> > > **Need actionable suggestions**
> > >
> > > Dear reviewer, thank you for your time and response.
> > >
> > > However, we find that your suggestions are somehow unactionable. In the first place, though FIT has already surpassed all previous baselines, the performance of it in the harder knowledge graph can not sky-rocket to nearly 90% like in FB15k and this result is consistent with many observations in previous work.
> > >
> > > Regarding your concerns about the proof, the tree-form query is formally defined by the Definition 8, therefore mathematically characterizes all its possibilities. If you have strong concerns about that, perhaps you can raise a counter-example or check our proof as it is already included in our appendix.
> > >
> > > We thank you again for your suggestions and looking forward to your reply.

---

> ### Comment · Reviewer_sXg1 · 2023-11-19
> **Acknowledgement**
>
> I accept and appreciate the clarifications

---

### Meta-Review · Area_Chair_ufPW · 2023-12-09

**Metareview:**

In this paper, the authors reviewed the limitations of current query embedding methods and extended the scope of complex query answering to the whole family of $EFO_1$ formulas. They also developed a new dataset containing ten formulas as well as analyzed the new difficulties coming with the new formulas. Finally, based on strict fuzzy logic theory, the authors present a new neural-symbolic method, FIT, which can fine-tune pretrained neural link predictor to infer arbitrary $EFO_1$ formula and outperform existing methods significantly.

The paper is well-written and easy to follow. The idea of the approach is interesting and the experimental results are good. As all the reviewers are willing to accept this paper, I also recommend accepting it.

**Justification For Why Not Higher Score:**

The technical contribution of the paper is somewhat limited

**Justification For Why Not Lower Score:**

The paper is good and all the reviewers are willing to accept it

---

### Decision · Program_Chairs · 2024-01-16

Accept (poster)